# FlowerTune: A Cross-Domain Benchmark for Federated Fine-Tuning of Large Language Models

**Yan Gao[1,2,]***, **Massimo Roberto Scamarcia[3], Javier Fernandez-Marques[1,2], Mohammad Naseri[1], Chong Shen Ng[1], Dimitris Stripelis[1], Zexi Li[2,4], Tao Shen[4], Jiamu Bai[5], Daoyuan Chen[6], Zikai Zhang[7], Rui Hu[7], InSeo Song[8], Lee KangYoon[8], Hong Jia[9], Ting Dang[10], Junyan Wang[11], Zheyuan Liu[11], Daniel Janes Beutel[1], Lingjuan Lyu[12], Nicholas D. Lane[1,2]**

[1]Flower Labs, [2]University of Cambridge, [3]ethicalabs.ai, [4]Zhejiang University,
[5]Penn State University, [6]Alibaba Group, [7]University of Nevada, Reno, [8]Gachon University,
[9]The University of Auckland, [10]The University of Melbourne, [11]The University of Adelaide, [12]Sony AI

## Abstract

Large Language Models (LLMs) have achieved state-of-the-art results across diverse domains, yet their development remains reliant on vast amounts of publicly available data, raising concerns about data scarcity and the lack of access to domain-specific, sensitive information. Federated Learning (FL) presents a compelling framework to address these challenges by enabling decentralized fine-tuning on pre-trained LLMs without sharing raw data. However, the compatibility and performance of pre-trained LLMs in FL settings remain largely under explored. We introduce the *FlowerTune LLM Leaderboard*, a *first-of-its-kind* benchmarking suite designed to evaluate federated fine-tuning of LLMs across four diverse domains: general NLP, finance, medical, and coding. Each domain includes federated instruction-tuning datasets and domain-specific evaluation metrics. Our results, obtained through a collaborative, open-source and community-driven approach, provide the first comprehensive comparison across 26 pre-trained LLMs with different aggregation and fine-tuning strategies under federated settings, offering actionable insights into model performance, resource constraints, and domain adaptation. This work lays the foundation for developing privacy-preserving, domain-specialized LLMs for real-world applications.

## 1 Introduction

Large language models (LLMs) [1, 2, 7, 24, 58, 66] have exhibited remarkable performance across a broad spectrum of machine learning tasks and domains, including general natural language processing (NLP), medical question answering [53, 55], financial sentiment analysis [67], and code generation [30, 31]. These capabilities are typically attained through supervised fine-tuning applied to large-scale pre-trained models [21, 26, 30].

Despite their success, the current paradigm of LLM development heavily depends on large volumes of publicly available data [1, 24, 58]. While it is widely acknowledged that increased data volume typically enhances model performance, recent studies have raised concerns that the supply of high-quality public data may be exhausted within a few years [56, 59]. Moreover, there is an increasing demand for specialized LLMs that can integrate domain-specific knowledge not readily accessible in publicly available web-based corpora—particularly in sensitive domains such as healthcare and finance. However, accessing data stored within relevant institutions and organizations raises substan-

---

*Corresponding author: yan@flower.ai

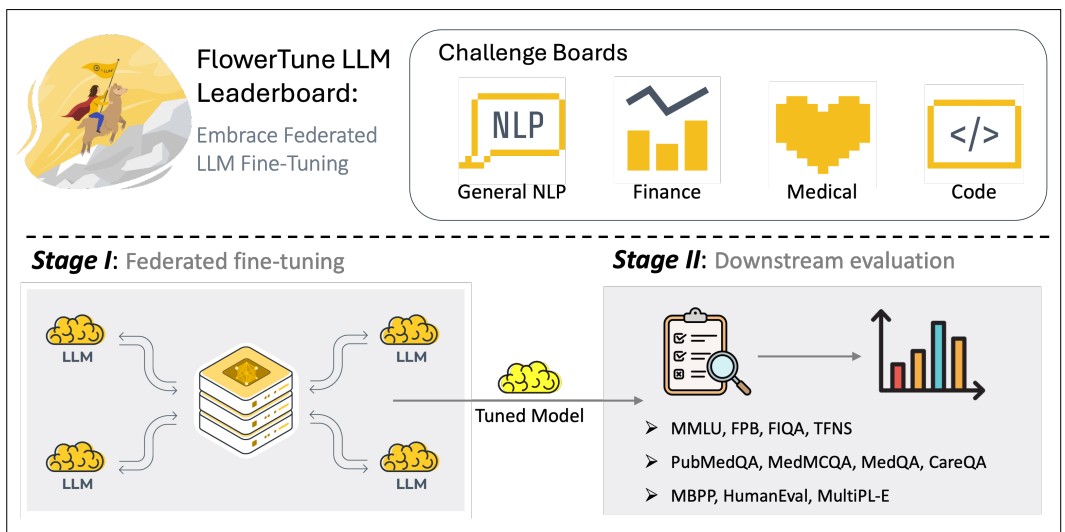

Figure 1: **Overview of the FlowerTune LLM Leaderboard.** This leaderboard provides four challenges covering: general NLP, finance, medical, and coding. After selecting a challenge, participants can initiate federated fine-tuning using a provided template tailored to the specific scenario. The template is model-agnostic and supports various pre-trained base models, fine-tuning strategies, and aggregation algorithms, enabling flexible adaptation. Upon completion of training, the resulting global LLM is evaluated using domain-specific metrics, with scores reported to reflect the performance and quality of the tuned model.

tial privacy and security concerns, and is further complicated by challenges related to data storage and transfer, and communication infrastructure.

LLM fine-tuning via federated learning (FL) offers a promising solution to address these challenges [36, 69, 70]. FL enables collaborative model optimization by facilitating access to a wider range of sensitive datasets without the need to share raw data. This approach holds significant potential for the development of more fairness-preserving [43, 54] and domain-adapted LLMs. Recent studies [36, 69, 70] have begun to explore FL algorithms and fine-tuning strategies on selected models within this context. With rapid advancements and increasing accessibility of LLM pre-training, a broad array of pre-trained models are now available for downstream fine-tuning. However, their suitability for deployment in FL environments remains largely unexplored. Several critical questions persist: How effectively do these models perform under federated settings using existing aggregation and fine-tuning strategies? Are they compatible with the prevalent resource constraints in FL scenarios, such as communication overhead and memory limitations? And can FL offer a practical pathway for training more domain-specialized LLMs?

To systematically address these open challenges and deepen understanding, we introduce the FlowerTune LLM Leaderboard [2], a benchmarking initiative built on the Flower Platform [13], designed to evaluate the performance of various pre-trained LLMs under various federated fine-tuning scenarios across multiple domains. The leaderboard focuses on four high-impact application areas involving sensitive or private data: general NLP, finance, medical, and coding. Each domain includes a dedicated federated dataset for instruction tuning, along with domain-specific evaluation metrics. By establishing baseline results across a range of models and domains in an open-source, community-driven framework, the FlowerTune LLM Leaderboard has attracted an increasing number of submissions from both academic and industry contributors. Building upon these contributions, we conduct a first-of-its-kind comprehensive set of fine-tuning experiments to benchmark 26 base models under a unified FL setting. This enables valuable insights into the feasibility and effectiveness of deploying federated fine-tuning for LLMs, and would further assist in accelerating the development of more inclusive, privacy-preserving, and domain-specialized language models for real-world applications.

---

[2]https://flower.ai/benchmarks/llm-leaderboard

Table 1: Statistical summary of the datasets and their splits for federated fine-tuning in FlowerTune LLM Leaderboard. The average length of instruction and output is measured in characters.

| Challenges | Fine-tuning dataset | Total # samples | # clients | $len$(instruction) | $len$(output) |
|---|---|---|---|---|---|
| General NLP | alpaca-gpt4 [50] | 52 K | 20 | 60 | 677 |
| Finance | fingpt-sentiment-train [67] | 76.8 K | 50 | 112 | 9 |
| Medical | medical-flashcards [25] | 34 K | 20 | 92 | 349 |
| Code | code-alpaca-20k [16] | 20 K | 10 | 74 | 197 |

Table 2: Summary of the evaluation datasets used in FlowerTune LLM Leaderboard. MQA represents multiple-choice question answering.

| Challenges | Evaluation datasets | Task types | Evaluation metrics |
|---|---|---|---|
| General NLP | MMLU [27] (STEM, Humanities, Social Sciences) | MQA | Accuracy |
| Finance | FPB [45], FIQA [44], TFNS [73] | Classification | Accuracy |
| Medical | PubMedQA [33], MedMCQA [48], MedQA [32], CareQA [8] | MQA | Accuracy |
| Code | MBPP [9], HumanEval [17], MultiPL-E (JS, C++) [15] | Code generation | Pass@1 |

## 2 FlowerTune LLM Leaderboard

### 2.1 Overview

The FlowerTune LLM Leaderboard is a benchmarking initiative that provides tools and standardized baselines for federated fine-tuning and evaluation of LLMs. It aims to facilitate reproducible research and community-driven exploration in this emerging, yet increasingly studied area. For reproducibility, this leaderboard includes two end-to-end pipelines for LLM federated fine-tuning and evaluation, respectively, involving four sensitive data domains: general NLP, finance, medical, and coding. More domains are also planned for future inclusion to further broaden its scope (Figure 1).

### 2.2 Federated fine-tuning

To emulate data distributions that could be expected across institutions such as medical, financial, and educational organizations, we carefully select four domain-specific public datasets to support the respective challenges. Specifically, we use alpaca-gpt4 [3] [50] for the general NLP domain, fingpt-sentiment-train [4] [67] for finance, medical-flashcards [5] [25] for the medical domain, and code-alpaca-20k [6] [16] for coding. Each dataset includes domain-specific instruction prompts paired with corresponding answers, designed to train an LLM to function as an assistant within its respective domain. We employ Flower Datasets [37] to partition each dataset into multiple shards of approximately equal size, simulating the data distribution across institutions in FL environments. A detailed statistical summary of the datasets and their corresponding splits is presented in Table 1, while representative example samples from each dataset are provided in the Appendix C. Additionally, a quantitative analysis of the client-level data distribution is provided in Appendix A.1. Although the number of samples is relatively balanced across clients within each domain, semantic similarity metrics reveal notable differences among clients, particularly in the general NLP, medical, and code domains.

After selecting a challenge, participants can implement their own federated methods and LLM models using the provided template. This template is model-agnostic and readily adaptable to various fine-tuning techniques and aggregation algorithms. Furthermore, the FlowerTune framework includes built-in tools for measuring key FL system metrics, such as communication overhead and memory usage. A detailed description of the participation process is provided in Appendix D.

### 2.3 Downstream evaluation

For each domain, we provide a corresponding evaluation pipeline that includes domain-specific evaluation datasets (Table 2). Specifically, in the general NLP challenge, the MMLU dataset [27]

---

[3] https://huggingface.co/datasets/flwrlabs/alpaca-gpt4

[4] https://huggingface.co/datasets/flwrlabs/fingpt-sentiment-train

[5] https://huggingface.co/datasets/flwrlabs/medical-meadow-medical-flashcards

[6] https://huggingface.co/datasets/flwrlabs/code-alpaca-20k

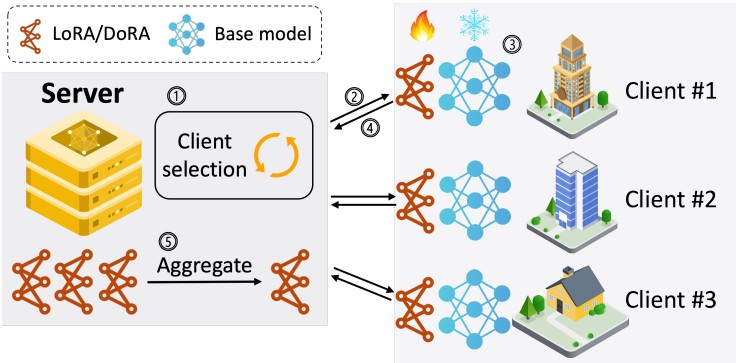

Figure 2: **Illustration of the federated LLM fine-tuning process.** (1) Initialization of LoRA/DoRA adapters and client selection on the server; (2) transmission of adapter parameters to the selected clients; (3) local adapter fine-tuning with the base model frozen; (4) transmission of updated adapter parameters back to the server; (5) aggregation of adapter parameters. This process is repeated in each subsequent FL round.

is used for evaluation. MMLU consists of multiple-choice questions covering a broad range of subjects across STEM, humanities, and social sciences. The evaluation metric is accuracy, reflecting the performance of the tested LLM. For the finance challenge, evaluation is conducted using a sentiment classification pipeline on financial reports and social media posts. Three datasets are utilized: FPB [45], FIQA [44], and TFNS [73]. Model performance is assessed using accuracy as the evaluation metric. In the medical domain, we leverage four datasets—PubMedQA [33], MedMCQA [48], MedQA [32], and CareQA [8]—to evaluate the performance of the tuned LLM as a medical assistant. Each dataset consists of multiple-choice questions covering various medical topics. Evaluation is based on accuracy, reflecting the model's ability to select the correct answers. For the coding challenge, we select MBPP [9], HumanEval [17], and MultiPL-E (JavaScript, C++) [15] as evaluation datasets. These datasets are specifically designed to assess code generation capabilities, and the pass@1 score is used as the evaluation metric, measuring the proportion of correctly generated programs on the first attempt.

The federated fine-tuned LLMs are evaluated in a zero-shot fashion, a challenging yet realistic scenario in which the model does not see any data samples from the evaluation domain during the fine-tuning phase. All models are evaluated under identical configurations to ensure a fair and consistent comparison.

## 3 Experimental settings

The FlowerTune LLM Leaderboard has attracted significant interest from the community, as evidenced by a growing number of submissions. These submissions, together with the benchmarking code [7] and pipelines provided by FlowerTune, were instructive in allowing us to systematically conduct a *first-of-its-kind* comprehensive set of fine-tuning experiments to benchmark various base models within a unified FL setting. In addition to investigating different base models, we also explore the impact of diverse aggregation algorithms and fine-tuning strategies. Detailed descriptions of the experimental settings are provided in the following sections.

### 3.1 Base model selection

Unlike traditional data centers, institutions participating in FL environments often lack access to large-scale computational resources, such as high-performance GPUs. As a result, hosting extremely large models is generally impractical. Taking these constraints into account—and aiming to enable effective federated fine-tuning—we focus on LLMs with fewer than 14 billion parameters.

In total, we select 26 base models with parameter sizes ranging from 135 million to 14 billion. These models are categorized into two groups: (1) pre-trained base models without further fine-tuning, and

---

[7] https://github.com/yan-gao-GY/flowertune-benchmark

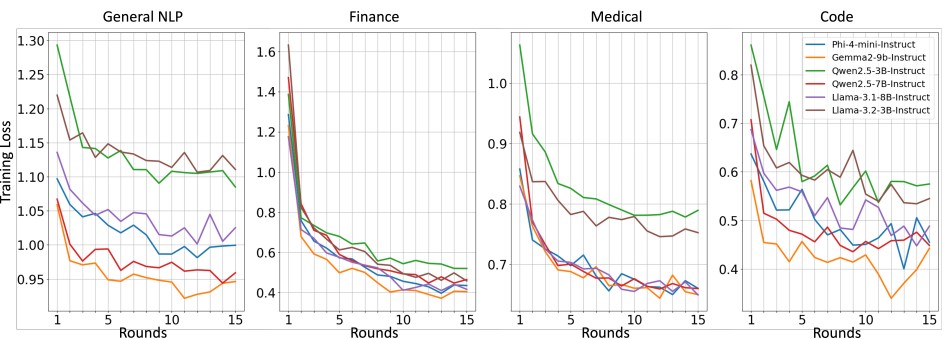

Figure 3: **Average training loss over FL rounds with 6 selected models on four challenges.** The training loss exhibits a consistent downward trend across all tasks, with larger fluctuations observed in the coding challenge.

(2) models that have been fine-tuned after pre-training by their respective providers, denoted with the suffix "-Instruct". Details of the base models are presented in Appendix A.2.

## 3.2 Federated learning settings

In the benchmarking experiments of different base models, a unified FL configuration is adopted across all challenges. Specifically, 20% of clients are selected per FL round, resulting in 4/20, 10/50, 4/20, and 2/10 clients for general NLP, finance, medical, and coding challenges, respectively (Table 1). Fine-tuning is conducted over 15 rounds for all tasks. During training, two system-level metrics are monitored: communication cost and VRAM usage. Communication cost is measured as the total bi-directional data transmitted between the server and selected clients across all FL rounds, while VRAM usage is recorded per client on a single GPU. The entire federated fine-tuning process is illustrated in Figure 2.

In the investigation of FL strategies, we select four popular algorithms: (1) **FedAvg** [46]: Federated Averaging (FedAvg) aggregates local model updates by computing their weighted average based on client data size. This method serves as a foundational method for many FL algorithms. (2) **Fed-Prox** [40]: Federated Proximal (FedProx) extends FedAvg by introducing a proximal term to the local objective, which helps stabilize training when client data is non-IID by discouraging local models from drifting too far from the global model. (3) **FedAvgM** [28]: FedAvg with Momentum (FedAvgM) enhances FedAvg by incorporating server-side momentum during aggregation, which accelerates convergence and improves performance in heterogeneous environments. (4) **FlexLoRA** [10]: FlexLoRA is a communication-efficient FL method that integrates Low-Rank Adaptation (LoRA) into local training and employs adaptive aggregation to fuse updates, which improves scalability and personalization.

## 3.3 Local fine-tuning hyper-parameters

During the local fine-tuning phase, we adopt parameter-efficient fine-tuning (PEFT) techniques [21, 26], which substantially reduce computational requirements and accelerate the training process. Specifically, we investigate the following two popular PEFT methods in our federated LLM fine-tuning pipeline: (1) **LoRA** [29] introduces a PEFT method by injecting trainable low-rank matrices into pre-trained model weights, enabling a significant reduction in trainable parameters and computational cost without additional inference latency. (2) **DoRA** [41] builds upon LoRA by decomposing weights into magnitude and direction components, applying LoRA specifically to the directional part to better emulate the learning capacity of full fine-tuning while preserving efficiency. Both methods are applied with quantization, resulting in a QLoRA-style training approach [20].

In the local tuning process, the base model remains frozen, while only the LoRA/DoRA adapters are trained and communicated between clients and the central server. We employ DoRA while benchmarking various base models and additionally compare its performance against LoRA on selected models. Additionally, we integrate FlashAttention-2 [19] to further enhance training speed and computational efficiency. All experiments are conducted on NVIDIA A100 SXM4 (80 GB) GPUs. The complete set of training hyper-parameters is summarized in the Appendix A.3.

Table 3: **Comparison of different base models (Instruct version) federated fine-tuned on the General NLP challenge.** The accuracy values (%) are reported on different downstream tasks. Comm. represents the total communication costs over FL fine-tuning, while Mem. stands for the VRAM costs per client. The highest performance and lowest system overhead are indicated in **bold**, while the second-best results are underlined.

| Models | STEM (%) | Social Sciences (%) | Humanities (%) | Average (%) | Comm. (GB) | Mem. (GB) |
|---|---|---|---|---|---|---|
| Mistral-7B-Instruct-v0.3 | 14.91 | 32.79 | 33.75 | 27.15 | 12.31 | 25.28 |
| Gemma2-9B-Instruct | 40.34 | 71.43 | 47.31 | 53.03 | 15.52 | 59.49 |
| Phi-4-mini-Instruct (3.8B) | **56.90** | 75.76 | 57.79 | 63.48 | 21.03 | 48.09 |
| Llama3.2-1B-Instruct | 0.54 | 1.07 | 1.49 | 1.03 | 3.31 | 27.53 |
| Llama3.2-3B-Instruct | 14.91 | 32.21 | 27.65 | 24.92 | 7.05 | 32.68 |
| Llama3.1-8B-Instruct | 52.87 | 69.65 | 50.01 | 57.51 | 8.10 | 45.85 |
| Qwen2-0.5B-Instruct | 20.20 | 27.46 | 25.21 | 24.29 | 2.61 | 26.67 |
| Qwen2.5-1.5B-Instruct | 41.45 | 63.67 | 49.93 | 51.68 | 5.48 | 31.97 |
| Qwen2.5-3B-Instruct | 34.22 | 49.82 | 34.20 | 39.41 | 8.90 | 34.74 |
| Qwen2.5-7B-Instruct | 55.47 | **78.23** | **59.81** | **64.50** | 11.99 | 49.31 |
| SmolLM2-135M-Instruct | 19.79 | 20.38 | 21.98 | 20.71 | **1.42** | **9.06** |
| SmolLM2-360M-Instruct | 14.53 | 17.48 | 19.51 | 17.17 | 2.52 | 11.28 |
| SmolLM2-1.7B-Instruct | 5.17 | 9.33 | 9.73 | 8.08 | 4.97 | 15.17 |

Table 4: **Comparison of different base models (Instruct version) federated fine-tuned on the Finance challenge.** The accuracy values (%) are reported on different downstream tasks. Comm. represents the total communication costs over FL fine-tuning, while Mem. stands for the VRAM costs per client. The highest performance and lowest system overhead are indicated in **bold**, while the second-best results are underlined.

| Models | FPB (%) | FIQA (%) | TFNS (%) | Average (%) | Comm. (GB) | Mem. (GB) |
|---|---|---|---|---|---|---|
| Mistral-7B-Instruct-v0.3 | 76.98 | 75.99 | 55.82 | 69.60 | 30.77 | 17.26 |
| Gemma2-9B-Instruct | **84.65** | **83.22** | **84.76** | **84.21** | 38.80 | 39.39 |
| Phi-4-mini-Instruct (3.8B) | 44.80 | 52.30 | 27.85 | 41.65 | 52.56 | 23.16 |
| Llama3.2-1B-Instruct | 55.94 | 57.89 | 51.84 | 55.23 | 8.28 | 11.04 |
| Llama3.2-3B-Instruct | 45.46 | 64.80 | 38.07 | 49.44 | 17.63 | 16.21 |
| Llama3.1-8B-Instruct | 40.18 | 64.47 | 32.79 | 45.81 | 30.77 | 27.55 |
| Qwen2-0.5B-Instruct | 37.38 | 47.37 | 37.81 | 40.85 | 6.54 | 10.78 |
| Qwen2.5-1.5B-Instruct | 32.01 | 59.87 | 22.07 | 37.98 | 13.69 | 14.58 |
| Qwen2.5-3B-Instruct | 29.29 | 62.50 | 21.61 | 37.80 | 22.26 | 17.21 |
| Qwen2.5-7B-Instruct | 29.70 | 60.53 | 21.36 | 37.20 | 29.97 | 28.50 |
| SmolLM2-135M-Instruct | 30.53 | 59.21 | 24.16 | 37.97 | **3.55** | **4.89** |
| SmolLM2-360M-Instruct | 27.89 | 58.55 | 24.41 | 36.95 | 6.30 | 5.55 |
| SmolLM2-1.7B-Instruct | 52.97 | 33.88 | 59.97 | 48.94 | 12.42 | 9.51 |

# 4 Experimental results

This section presents the results of three core experimental investigations: (1) benchmarking the performance of various base models under a unified FL setup, (2) analyzing system performance during federated fine-tuning, and (3) evaluating the impact of different aggregation and fine-tuning techniques. It is important to note that this benchmark aims to provide a standardized testbed of different base models under FL settings, serving as a reference for future studies. All experiments are conducted using a unified configuration without task-specific or model-specific hyper-parameter tuning. More results with particular hyper-parameter tuning are available on the FlowerTune LLM Leaderboard [2].

## 4.1 Evaluation performance analysis

We examine both instruct and non-instruct base models within federated environments. Results for instruct models are presented in Tables 3–6, while those for non-instruct models are shown in Tables 12–15. Overall, instruct models demonstrate more stable and consistently higher performance across all four domains compared to their non-instruct counterparts. A detailed analysis of the instruct models is provided in the following sections, with the analysis of non-instruct models is included in the Appendix B.1.

Table 3 shows the evaluated performance on the **general NLP** challenge. First, the Qwen2.5-7B-Instruct model achieves the highest average accuracy across the three evaluation datasets, outperforming larger models such as Gemma2-9B-Instruct and LlaMA-3.1-8B-Instruct. Second, Phi-4-Mini-Instruct obtains the second-highest performance (63.48%) despite having a relatively small parameter count of 3.8 billion. Interestingly, Qwen2.5-1.5B-Instruct and SmolLM2-135M-Instruct achieve sur-

Table 5: **Comparison of different base models (Instruct version) federated fine-tuned on the Medical challenge.** The accuracy values (%) are reported on different downstream tasks. Comm. represents the total communication costs over FL fine-tuning, while Mem. stands for the VRAM costs per client. The highest performance and lowest system overhead are indicated in **bold**, while the second-best results are underlined.

| Models | PubMedQA (%) | MedMCQA (%) | MedQA (%) | CareQA (%) | Average (%) | Comm. (GB) | Mem. (GB) |
|---|---|---|---|---|---|---|---|
| Mistral-7B-Instruct-v0.3 | 63.80 | 18.55 | 39.83 | 28.39 | 37.64 | 12.31 | 20.71 |
| Gemma2-9B-Instruct | 67.60 | **54.46** | **57.34** | **69.58** | **62.25** | 15.52 | 53.40 |
| Phi-4-mini-Instruct (3.8B) | 13.60 | 42.12 | 47.60 | 56.08 | 39.85 | 21.03 | 34.38 |
| Llama3.2-1B-Instruct | 19.20 | 7.32 | 24.12 | 4.38 | 13.75 | 3.31 | 20.39 |
| Llama3.2-3B-Instruct | 30.20 | 1.65 | 6.99 | 4.39 | 10.81 | 7.05 | 23.96 |
| Llama3.1-8B-Instruct | **68.80** | 17.83 | 32.36 | 30.55 | 37.39 | 12.31 | 36.24 |
| Qwen2-0.5B-Instruct | 56.80 | 8.70 | 13.75 | 8.15 | 21.85 | 2.61 | 18.21 |
| Qwen2.5-1.5B-Instruct | 48.80 | 41.33 | 42.58 | 50.51 | 45.80 | 5.48 | 22.29 |
| Qwen2.5-3B-Instruct | 65.20 | 39.30 | 34.25 | 45.37 | 46.03 | 8.90 | 24.83 |
| Qwen2.5-7B-Instruct | 60.80 | 49.77 | 56.87 | 64.06 | 57.88 | 11.99 | 39.05 |
| SmolLM2-135M-Instruct | 54.00 | 20.01 | 9.35 | 16.79 | 25.04 | **1.42** | **7.06** |
| SmolLM2-360M-Instruct | 40.20 | 7.75 | 3.46 | 10.96 | 15.59 | 2.52 | 7.79 |
| SmolLM2-1.7B-Instruct | 39.20 | 6.60 | 11.78 | 11.44 | 17.26 | 4.97 | 11.82 |

Table 6: **Comparison of different base models (Instruct version) federated fine-tuned on the Coding challenge.** The pass@1 scores (%) are reported on different downstream tasks. Comm. represents the total communication costs over FL fine-tuning, while Mem. stands for the VRAM costs per client. The highest performance and lowest system overhead are indicated in **bold**, while the second-best results are underlined.

| Models | MBPP (%) | HumanEval (%) | MultiPL-E (JS) (%) | MultiPL-E (C++) (%) | Average (%) | Comm. (GB) | Mem. (GB) |
|---|---|---|---|---|---|---|---|
| Mistral-7B-Instruct-v0.3 | 40.40 | 37.80 | 43.48 | 33.54 | 38.81 | 6.15 | 22.65 |
| Gemma2-9B-Instruct | **53.40** | 58.54 | 54.04 | 47.20 | **53.29** | 7.76 | 58.05 |
| Phi-4-mini-Instruct (3.8B) | 47.80 | **60.98** | 54.04 | 38.51 | 50.33 | 10.51 | 38.00 |
| Llama3.2-1B-Instruct | 34.80 | 35.98 | 22.98 | 18.63 | 28.10 | 1.66 | 21.29 |
| Llama3.2-3B-Instruct | 43.80 | 54.88 | 46.58 | 34.16 | 44.86 | 3.53 | 26.79 |
| Llama3.1-8B-Instruct | 50.40 | 59.76 | **58.39** | 44.10 | 53.16 | 6.15 | 37.41 |
| Qwen2-0.5B-Instruct | 13.20 | 14.63 | 13.04 | 14.91 | 13.95 | 1.31 | 19.74 |
| Qwen2.5-1.5B-Instruct | 22.80 | 6.10 | 8.07 | 24.84 | 15.45 | 2.74 | 24.09 |
| Qwen2.5-3B-Instruct | 40.20 | 22.56 | 6.83 | 37.89 | 26.87 | 4.45 | 27.41 |
| Qwen2.5-7B-Instruct | 48.40 | 18.90 | 9.94 | **50.31** | 31.89 | 5.99 | 40.24 |
| SmolLM2-135M-Instruct | 8.40 | 7.93 | 4.97 | 4.97 | 6.57 | **0.71** | **8.84** |
| SmolLM2-360M-Instruct | 25.00 | 18.29 | 13.04 | 10.56 | 16.72 | 1.26 | 9.09 |
| SmolLM2-1.7B-Instruct | 34.60 | 28.05 | 21.12 | 24.84 | 27.15 | 2.48 | 12.42 |

prisingly competitive performance, even surpassing some of their larger counterparts. This suggests their potential suitability for more resource-constrained FL scenarios in general NLP fine-tuning.

In the **finance** challenge (Table 4), Gemma2-9B-Instruct achieves the best average performance (84.21%), largely attributable to its greater capacity stemming from the largest number of parameters among the evaluated models. Mid-sized models (e.g., Llama3.2-3B, Qwen2.5-3B) obtain performance comparable to their larger counterparts (e.g., Llama3.1-8B, Qwen2.5-7B), a trend that is also reflected in their training loss trajectories (Figure 3). Smaller models such as LlaMA-3.2-1B-Instruct, Qwen2-0.5B-Instruct, and SmolLM2-135M-Instruct perform better in this challenge. This could be partially due to the relative simplicity of the classification task, which allows even lightweight models to learn effectively.

In the **medical** challenge (Table 5), Gemma2-9B-Instruct again achieves the highest average accuracy of 62.25% across the four evaluation datasets. Note that while most federated fine-tuned models perform well on specific datasets, they lack generalization across all tasks within the medical domain. Overall, the Qwen model family consistently outperforms the LlaMA models of comparable size in this medical QA evaluation, suggesting that LlaMA models may require more careful fine-tuning under FL settings. Notably, SmolLM2-135M-Instruct obtains an acceptable average accuracy of 25.04%, outperforming some larger models and demonstrating its potential in resource-constrained federated environments.

Table 6 presents the evaluation results for the **coding** challenge. Gemma2-9B-Instruct (53.29%) achieved the highest performance followed closely by LlaMA-3.1-8B-Instruct (53.16%), whereas smaller models with fewer than 3 billion parameters generally underperform. This suggests that code generation tasks may inherently require the capacity provided by larger models to achieve strong performance. Phi-4-Mini-Instruct stands out by achieving an average accuracy of 50.33%, demonstrating strong performance relative to its modest size of 3.8 billion parameters.

In summary, larger models generally exhibit superior performance across all challenges. However, the nature of the task—such as question answering, classification, or code generation—should be

Table 7: Comparison of **aggregation methods** using the Qwen2.5-7B base model across four challenges. DoRA is used in the local fine-tuning. The highest performance is indicated in **bold**.

| | General NLP | | | | Medical | | | | |
|---|---|---|---|---|---|---|---|---|---|
| Methods | STEM | Social Sciences | Humanities | Avg | PubMedQA | MedMCQA | MedQA | CareQA | Avg |
| FedAvg | **41.55** | 52.62 | 34.20 | 42.79 | 33.40 | **17.62** | 18.85 | **30.21** | **25.02** |
| FedProx | 41.52 | **52.78** | 34.45 | **42.92** | 36.20 | 15.18 | 15.95 | 26.31 | 23.41 |
| FedAvgM | 39.80 | 49.56 | **35.30** | 41.56 | 36.80 | 15.61 | 14.77 | 24.98 | 23.04 |
| FlexLoRA | 34.82 | 46.05 | 30.31 | 37.06 | **42.40** | 13.08 | **19.01** | 16.42 | 22.73 |

| | Finance | | | | Code | | | | |
|---|---|---|---|---|---|---|---|---|---|
| Methods | FPB | FIQA | TFNS | Avg | MBPP | HumanEval | MultiPL-E (JS) | MultiPL-E (C++) | Avg |
| FedAvg | **71.86** | 52.30 | 74.20 | 66.12 | 56.60 | 23.17 | 52.17 | 49.07 | 45.25 |
| FedProx | 71.62 | 54.61 | 74.08 | 66.77 | 55.40 | 21.95 | 52.80 | 48.45 | 44.65 |
| FedAvgM | 73.76 | **56.91** | **74.25** | **68.31** | **59.60** | **25.61** | **53.42** | 48.45 | **46.77** |
| FlexLoRA | 70.96 | 52.96 | 72.28 | 65.40 | 57.40 | 15.85 | 49.69 | **50.93** | 43.47 |

Table 8: Comparison of two **fine-tuning techniques** using the Qwen2.5-7B base model across four challenges. FedAvg is used for aggregation. "SS" refers to Social Sciences. For the medical challenge, "PMQA", "MMCQA", "MQA" and "CQA" correspond to PubMedQA, MedMCQA, MedQA, and CareQA, respectively. For the coding challenge, "HE", "M-JS" and "M-C++" represent HumanEval, MultiPL-E (JS), and MultiPL-E (C++). Communication (Comm.) and VRAM (Mem.) costs are reported in gigabytes.

| | General NLP | | | | | | Medical | | | | | | |
|---|---|---|---|---|---|---|---|---|---|---|---|---|---|
| Methods | STEM | SS | Humanities | Avg | Comm. | Mem. | PMQA | MMCQA | MQA | CQA | Avg | Comm. | Mem. |
| LoRA | 38.47 | 48.33 | 33.75 | 40.18 | 11.90 | 46.07 | 37.80 | 16.69 | 18.93 | 28.13 | 25.39 | 11.90 | 34.88 |
| DoRA | 41.55 | 52.62 | 34.20 | 42.79 | 11.99 | 49.23 | 33.40 | 17.62 | 18.85 | 30.21 | 25.02 | 11.99 | 39.53 |

| | Finance | | | | | | Code | | | | | | |
|---|---|---|---|---|---|---|---|---|---|---|---|---|---|
| Methods | FPB | FIQA | TFNS | Avg | Comm. | Mem. | MBPP | HE | M-JS | M-C++ | Avg | Comm. | Mem. |
| LoRA | 73.43 | 55.92 | 74.66 | 68.01 | 29.74 | 27.91 | 56.00 | 29.27 | 55.90 | 50.31 | 47.87 | 5.95 | 38.88 |
| DoRA | 71.86 | 52.30 | 74.20 | 66.12 | 29.97 | 28.10 | 56.60 | 23.17 | 52.17 | 49.07 | 45.25 | 5.99 | 39.36 |

carefully considered when selecting models. For instance, smaller models perform adequately on the finance classification task. While further advancements will benefit from algorithms specifically tailored to federated LLM fine-tuning, this benchmark offers a solid foundation and practical reference for selecting appropriate base models based on task requirements in federated learning settings.

## 4.2 System performance analysis

Tables 3–6 also report system-level performance metrics for different base models during federated fine-tuning. Overall, both communication costs and VRAM usage remain modest, primarily due to the adoption of the DoRA strategy. Communication overhead is influenced by three main factors: the size of the DoRA/LoRA adapters, the number of clients selected per round, and the total number of federated training rounds. For example, the finance task incurs higher communication costs than other tasks, solely due to the selection of more clients per round under otherwise identical settings. Controlling communication overhead is crucial in FL environments, particularly in scenarios with limited network bandwidth (see Appendix B.2).

VRAM consumption is influenced by several factors, including the architecture and size of the base model, the adapter size, and the input sequence length determined by the training dataset. Memory usage is often regarded as a primary bottleneck in federated on-device training. Experimental results demonstrate that the memory footprint of all evaluated models using FlowerTune tools remains below 80 GB—sufficient to fit within a single modern A100/H100 GPU, or across two mid-tier GPUs. In particular, larger models (e.g., Gemma2-9B-Instruct) demonstrate superior performance but incur higher memory consumption. Conversely, smaller models—such as those in the SmolLM2 family—achieve acceptable performance on specific challenges while requiring approximately 10 GB or less of VRAM. This enables the training of these smaller models on edge devices, such as an NVIDIA Jetson board, MacBook, or Raspberry Pi with 16 GB of RAM, when considering memory consumption alone. However, computational speed represents another critical factor, which can only be accurately assessed on actual edge devices and is generally considerably slower than GPU-based training. These trade-offs between performance and resource efficiency should be carefully considered when deploying models in real-world applications (see Appendix B.2).

### 4.3 Aggregation and fine-tuning techniques analysis

We evaluate the performance of four FL strategies using the Qwen2.5-7B base model, as presented in Table 7. Overall, the strategies yield comparable results, with only marginal differences observed across the various challenges. Specifically, based on average scores, FedProx achieves the best performance in the general NLP task, FedAvg performs best in the medical task, and FedAvgM outperforms others in both the finance and coding tasks. However, when examining individual evaluation datasets with a specific domain, no single strategy consistently outperforms the others across all datasets. These findings highlight the need for future research to develop more specialized aggregation algorithms tailored to federated LLM fine-tuning.

Table 8 presents a comparison of LoRA and DoRA in federated fine-tuning using the Qwen2.5-7B base model across four challenges. The results indicate that LoRA achieves slightly better average performance than DoRA in the finance and coding tasks, while both methods yield nearly identical accuracy in the medical task. In contrast, DoRA performs marginally better in the general NLP task. In terms of communication and VRAM costs, both LoRA and DoRA demonstrate efficient resource usage within modest limits. Particularly, DoRA incurs slightly higher overhead in both dimensions, mainly due to the additional computation required for decomposing weights into magnitude and direction components.

In summary, off-the-shelf aggregation strategies and fine-tuning techniques yield relatively minor differences in performance. In contrast, the choice of base model has a substantially greater impact, with notable performance variations observed across different domains. These findings highlight the critical importance of base model selection for federated LLM fine-tuning.

## 5 Related work

### 5.1 LLM instruction tuning

Instruction tuning is a form of fine-tuning that aligns LLMs with human intent using instruction-response datasets. Unlike task-specific fine-tuning, it promotes generalization across diverse tasks by training models to follow natural language prompts [74]. InstructGPT demonstrated that human-written instructions improve model helpfulness and safety [47], while Self-Instruct reduced data collection costs by using LLMs to generate synthetic instructions [61]. This approach enabled large-scale datasets like Super-NaturalInstructions and inspired methods such as Evo-Instruct and Instruct-SkillMix [34, 60, 72]. Parameter-efficient fine-tuning (PEFT) methods, such as LoRA, address the high cost of fine-tuning by updating only small portions of model parameters [29]. This makes instruction tuning feasible in low-resource environments. Recent studies integrate LoRA into instruction workflows for improved scalability [12, 20], while advanced variants like ALoRA optimize performance by dynamically adjusting parameter allocation [42]. Beyond supervised tuning, reinforcement learning from human feedback (RLHF) aligns model behavior with human preferences, as seen in the training of ChatGPT [11]. While effective, RLHF faces challenges in reward modeling, feedback quality, and scalability [76, 39].

Recent work has explored efficient on-device fine-tuning of LLMs to address privacy and resource constraints. GSQ-Tuning [75] enables fully integer-based on-device LLM fine-tuning using group-shared exponents, while QEFT [38] improves efficiency by updating only sensitive weight columns in mixed precision. These methods extend PEFT methods like LoRA [29] and QLoRA [20] for resource-constrained settings, and are applicable to federated cross-device fine-tuning. While our study focuses on cross-silo fine-tuning, we acknowledge that recent advances in on-device fine-tuning offer promising directions that could further enhance privacy and efficiency in federated settings, and we leave their exploration to future work.

### 5.2 Federated LLM fine-tuning

Federated LLM fine-tuning facilitates privacy-preserving training across decentralized data sources. Several frameworks and benchmarks have been developed to support this paradigm. Open-FedLLM [70] introduces a general framework for instruction tuning and alignment of LLMs without sharing raw data. FederatedScope-LLM [36] provides a modular package supporting PEFT and automated benchmarking. FATE-LLM [22] focuses on industrial deployment, supporting both horizontal

and vertical federated settings. FedLLM-Bench [69] addresses the benchmarking gap by offering realistic datasets, fine-tuning methods, and evaluation pipelines to support empirical studies. To enrich training data, FedIT-U2S [68] transforms unstructured client data into instruction-response pairs, while FedDCA [62] augments domain coverage to improve model generalization.

Despite these advancements, federated LLM fine-tuning faces challenges such as resource constraints, system heterogeneity, personalization, and communication efficiency. Parameter-efficient fine-tuning methods, particularly LoRA-based techniques, have become central to addressing these issues. FedBiOT [64] addresses limited resources by enabling clients to tune lightweight adapters instead of full models. FedALT [14] improves LoRA performance in federated contexts by stabilizing client updates via adaptive training. Further improvements in LoRA are seen in FFA-LoRA [57], which freezes select LoRA matrices to reduce communication, and in pFedLoRA [71], which supports model heterogeneity using shared small adapters. FLoRA [63] introduces aggregation strategies for heterogeneous LoRA modules, while FDLoRA [52] deploys dual adapters for balancing global and personalized knowledge. FedRand [49] enhances privacy via randomized LoRA parameter updates, and LoRA-A2 [35] adapts LoRA structure under extreme heterogeneity.

Beyond adapter tuning, several approaches focus on client-specific optimization. FedMKT [23] enhances generalization by transferring knowledge between small and large models. FedBis [65] integrates user preference alignment through binary selectors. These methods reflect growing attention to personalization and robustness in federated LLM fine-tuning. However, existing work has not systematically examined the impact of diverse base models on federated fine-tuning, particularly in terms of downstream performance and system-level behavior—an essential but underexplored aspect of this emerging field.

## 6 Conclusion

As the development of LLMs continues to advance, challenges surrounding data availability, privacy, domain specificity, and the practicalities of data storage and communication infrastructure are becoming increasingly pronounced. Federated LLM fine-tuning provides a potential solution through collaborative model optimization on sensitive and institutionally siloed data without compromising privacy. To address the lack of standardized evaluation in this emerging area, we introduced the *FlowerTune LLM Leaderboard*, an open-source and community-driven benchmarking platform designed to assess the performance of pre-trained LLMs under federated fine-tuning across four critical application domains: general NLP, finance, medical, and code. By providing domain-specific datasets, evaluation protocols, and baseline results, our initiative establishes a foundation for systematic, reproducible research on federated LLM fine-tuning.

**Limitations and future work**. This study focuses primarily on cross-silo federated settings; future work may explore a broader range of scenarios, including more resource-constrained environments and more heterogeneous data distributions. Additionally, the experiments were conducted using a unified configuration following a systematic protocol but without extensive hyper-parameter tuning for each individual model-dataset pair. Performance could likely be improved through more careful hyper-parameter optimization. Furthermore, the experimental results indicate that several classical federated aggregation algorithms (e.g., FedAvg, FedProx) exhibit only marginal performance differences in the context of LLM adapter tuning. This finding highlights the need for the community to develop aggregation algorithms specifically tailored to this parameter space (i.e., low-rank adapters) to more effectively integrate knowledge from diverse clients, representing a promising direction for future research. We envision this work as a catalyst for developing more inclusive, secure, and domain-adapted language models, and invite the broader research community to contribute to and extend this benchmarking effort.

## Acknowledgments and Disclosure of Funding

We sincerely thank all the participants in the FlowerTune LLM Leaderboard from the community for their valuable contributions and feedback. We are especially grateful to the following teams and individuals: Abdelkareem Elkhateb, AI4EOSC Team, Alessandro Pinto, CAR@AIML, Daniel Hinjos García, Dongqi Cai, FL-finetune-JB-DC, Gachon Cognitive Computing Lab, Massimo R. Scamarcia, mHealth Lab, Nut Chukamphaeng, T-IoI@UNR, and ZJUDAI.

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

# A Detailed experimental settings

This section provides details on client-level data distribution (Section A.1), the base models (Section A.2), and hyper-parameter configurations (Section A.3) used in the experiments.

## A.1 Client-level data distribution

From Table 9, we observe that the number of samples is relatively balanced across clients within each domain – a realistic assumption for cross-institutional FL settings. Variation in instruction and response lengths arises primarily from the inherent properties of the source datasets. We did not partition client data based on class distributions, as the instruction-tuning datasets used in this study are unlabeled. However, the semantic similarity metrics reveal meaningful differences across clients in each domain, particularly in the general NLP, medical, and code domains, where similarity scores approach 0, indicating a notable degree of heterogeneity. The higher similarity values observed in the finance domain are likely inherent to the original dataset prior to splitting.

Table 9: Statistical summary of client-level data distribution for four challenges. "# of Samples" indicates the number of data points per client. "Instruct Avg Length" and "Response Avg Length" represent the average sequence lengths of instructions and responses, respectively, for each client. For "Semantic Similarity," we compute embeddings for each instruction–response pair using the pre-trained all-MiniLM-L6-v2 model and then average these embeddings for each client. After applying dimensionality reduction, we calculate the cosine similarity between each client and all other clients, reporting the average similarity value per client to illustrate semantic variation (with values closer to 1 indicating greater similarity).

| General NLP | |
|---|---|
| # of Samples | [2601, 2601, 2600, 2600, 2600, 2600, 2600, 2600, 2600, 2600, 2600, 2600, 2600, 2600, 2600, 2600, 2600, 2600, 2600, 2600] |
| Instruct Avg Length | [59, 59, 59, 60, 59, 59, 59, 59, 60, 59, 59, 60, 59, 59, 60, 58, 59, 60, 59, 59] |
| Response Avg Length | [668, 668, 663, 671, 680, 673, 676, 664, 686, 672, 688, 708, 646, 692, 688, 680, 689, 693, 671, 660] |
| Semantic Similarity | [-0.02, 0.04, 0.03, -0.13, -0.07, 0.03, 0.04, -0.14, -0.0, -0.0, -0.1, -0.14, 0.04, -0.05, -0.04, -0.14, -0.08, -0.14, -0.05, 0.04] |

| Finance | |
|---|---|
| # of Samples | [1536, 1536, 1536, 1536, 1536, 1536, 1536, 1536, 1536, 1536, 1536, 1536, 1536, 1536, 1536, 1536, 1536, 1536, 1536, 1536, 1536, 1536, 1535, 1535, 1535, 1535, 1535, 1535, 1535, 1535, 1535, 1535, 1535, 1535, 1535, 1535, 1535, 1535, 1535, 1535, 1535, 1535, 1535, 1535, 1535, 1535] |
| Instruct Avg Length | [224, 225, 223, 223, 224, 223, 225, 223, 220, 226, 226, 224, 223, 226, 228, 225, 225, 228, 225, 221, 225, 223, 225, 223, 221, 224, 224, 225, 226, 223, 226, 224, 226, 221, 226, 227, 223, 223, 220, 228, 229, 224, 227, 226, 224] |
| Response Avg Length | [9, 9, 9, 9, 9, 9, 9, 9, 9, 9, 9, 9, 9, 9, 9, 9, 9, 9, 9, 9, 9, 9, 9, 9, 9, 9, 9, 9, 9, 9, 9, 9, 9, 9, 9, 9, 9, 9, 9, 9, 9, 9, 9, 9, 9] |
| Semantic Similarity | [0.72, 0.47, 0.29, 0.65, 0.51, 0.79, 0.79, 0.35, 0.33, 0.67, 0.78, 0.14, 0.79, 0.78, 0.76, 0.7, 0.7, 0.77, 0.71, 0.13, 0.78, 0.58, 0.72, 0.79, 0.64, 0.79, 0.65, 0.63, 0.46, 0.31, 0.79, 0.61, 0.69, 0.79, 0.79, 0.79, 0.77, 0.79, 0.24, 0.41, 0.77, 0.34, 0.77, 0.26, 0.74, 0.76, 0.79, 0.58, 0.77, 0.77] |

| Medical | |
|---|---|
| # of Samples | [1698, 1698, 1698, 1698, 1698, 1698, 1698, 1698, 1698, 1698, 1698, 1698, 1698, 1698, 1698, 1697, 1697, 1697, 1697, 1697] |
| Instruct Avg Length | [92, 92, 92, 90, 92, 93, 94, 91, 92, 91, 92, 91, 91, 92, 92, 92, 91, 93, 92, 92] |
| Response Avg Length | [346, 361, 346, 331, 340, 357, 362, 343, 343, 343, 353, 354, 355, 345, 351, 358, 341, 346, 344, 352] |
| Semantic Similarity | [-0.08, -0.04, -0.02, -0.08, -0.05, -0.03, -0.03, -0.03, -0.05, -0.07, -0.04, -0.07, -0.08, -0.03, -0.02, -0.06, -0.06, -0.09, -0.02, -0.08] |

| Code | |
|---|---|
| # of Samples | [2003, 2003, 2002, 2002, 2002, 2002, 2002, 2002, 2002, 2002] |
| Instruct Avg Length | [96, 96, 101, 97, 97, 99, 97, 98, 98, 98] |
| Response Avg Length | [199, 191, 193, 202, 193, 201, 200, 190, 196, 200] |
| Semantic Similarity | [0.03, -0.1, -0.24, -0.27, -0.05, -0.15, 0.05, 0.03, 0.04, -0.14] |

## A.2 Base model configurations

Table 10 presents detailed specifications of the base models used in the experiments, including the number of parameters and vocabulary size for each model.

## A.3 Hyper-parameter configurations

To ensure a fair comparison across different pre-trained base models, consistent hyper-parameters are applied to all tested LLMs. Table 11 outlines the local training hyper-parameter configurations used for base model benchmarking, as discussed in Section 4. For the Mistral 24B models, the DoRA rank and alpha values are adjusted to 16 and 32, respectively, to accelerate training and enable execution on a single GPU.

Table 10: Detailed information of the base models used in the experiments.

| Groups | Model name | # parameters | Vocabulary size (thousand) |
|---|---|---|---|
| Non-Instruct | Mistral-7B-v0.3 [5] | 7 B | 33 |
| | Mistral-Small-24B-Base-2501 [3] | 24 B | 131 |
| | Gemma-2-9B [58] | 9 B | 256 |
| | Phi-4 [1] | 14 B | 100 |
| | Llama-3.2-1B [24] | 1 B | 128 |
| | Llama-3.2-3B [24] | 3 B | 128 |
| | Llama-3.1-8B [24] | 8 B | 128 |
| | Qwen2-0.5B [18] | 0.5 B | 152 |
| | Qwen2.5-1.5B [66] | 1.5 B | 152 |
| | Qwen2.5-3B [66] | 3 B | 152 |
| | Qwen2.5-7B [66] | 7 B | 152 |
| | SmolLM2-135M [7] | 135 M | 49 |
| | SmolLM2-360M [7] | 360 M | 49 |
| | SmolLM2-1.7B [7] | 1.7 B | 49 |
| Instruct | Mistral-7B-Instruct-v0.3 [6] | 7 B | 33 |
| | Mistral-Small-24B-Instruct-2501 [4] | 24 B | 131 |
| | Gemma-2-9B-Instruct [51] | 9 B | 256 |
| | Phi-4-Mini-Instruct [2] | 3.8 B | 200 |
| | Llama-3.2-1B-Instruct [24] | 1 B | 128 |
| | Llama-3.2-3B-Instruct [24] | 3 B | 128 |
| | Llama-3.1-8B-Instruct [24] | 8 B | 128 |
| | Qwen2-0.5B-Instruct [18] | 0.5 B | 152 |
| | Qwen2.5-1.5B-Instruct [66] | 1.5 B | 152 |
| | Qwen2.5-3B-Instruct [66] | 3 B | 152 |
| | Qwen2.5-7B-Instruct [66] | 7 B | 152 |
| | SmolLM2-135M-Instruct [7] | 135 M | 49 |
| | SmolLM2-360M-Instruct [7] | 360 M | 49 |
| | SmolLM2-1.7B-Instruct [7] | 1.7 B | 49 |

# B    Additional experimental results

## B.1    Federated fine-tuning on non-instruct pre-trained base models

Tables 12–15 present the performance and system metrics of non-instruct pre-trained base models after federated fine-tuning across four challenges. Overall, these non-instruct base models underperform compared to their instruct-tuned counterparts, particularly in the case of smaller models, which may require more extensive fine-tuning to achieve competitive results. Gemma2-9B and Phi-4 demonstrate consistently strong performance across all domains, albeit with the highest communication and memory overhead. Notably, smaller models such as Qwen2.5-7B and Qwen2.5-3B achieve competitive results on the finance task, outperforming larger models in the 9B and 14B parameter range. In contrast, the Llama model family exhibits suboptimal performance on the medical challenge, suggesting a need for more tailored fine-tuning strategies in this domain.

## B.2    Pareto analysis of evaluation performance

To analyze the results from tables 12 − 15 from a computational trade-off perspective, we present the Pareto plots in figures 4 − 7 for the different base models (non-instruct version) when federated fine-tuned for the different tasks. When evaluating communication costs, the Qwen family of models stands out largely as the model family with the optimal trade-off. The Phi-4 model has a strong model performance (with the exception of the Finance challenge), but this is outweighed by the communication costs, which can vary up to a factor of 5. For performance versus VRAM evaluation, the results are mixed: there is strictly no optimal model in the General NLP challenge, whereas for

Table 11: Local training hyper-parameter configurations used for base model benchmarking. The values of DoRA rank and Alpha are adjusted to 16 and 32 when training Mistral 24B models.

| Hyper-parameters | Values |
|---|---|
| DoRA/LoRA rank (r) | 32 |
| DoRA/LoRA Alpha | 64 |
| Batch size | 16 |
| Optimizer | AdamW |
| Sequence length | 512 |
| Maximum number of steps | 10 |
| Accumulation steps | 1 |
| Maximum gradient norm | 1.0 |
| LR scheduler over rounds | Cosine annealing |
| LR scheduler over steps | Constant |
| Maximum LR | 5e-5 |
| Minimum LR | 1e-6 |
| Quantization | 4-bit |
| Precision | bf16 |

Table 12: **Comparison of different non-instruct base models federated fine-tuned on the General NLP challenge.** The accuracy values (%) are reported on different downstream tasks. Comm. represents the total communication costs over FL fine-tuning, while Mem. stands for the VRAM costs per client. The highest average performance is indicated in **bold**, while the second-highest is underlined.

| Models | STEM (%) | Social Sciences (%) | Humanities (%) | Average (%) | Comm. (GB) | Mem. (GB) |
|---|---|---|---|---|---|---|
| Mistral-7B-v0.3 | 3.39 | 12.97 | 25.80 | 14.05 | 12.31 | 25.08 |
| Gemma2-9B | 28.83 | 54.99 | 41.59 | 41.80 | 15.52 | 60.48 |
| Phi-4 (14B) | 37.87 | 72.67 | 46.99 | **52.51** | 50.77 | 57.70 |
| Llama3.2-1B | 3.36 | 4.58 | 2.23 | 3.39 | 3.31 | 29.08 |
| Llama3.2-3B | 1.21 | 1.43 | 0.79 | 1.14 | 7.05 | 32.97 |
| Llama3.1-8B | 0.13 | 0.62 | 0.23 | 0.33 | 8.10 | 46.58 |
| Qwen2-0.5B | 2.73 | 5.36 | 3.63 | 3.91 | 2.61 | 28.40 |
| Qwen2.5-1.5B | 21.12 | 21.68 | 24.19 | 22.33 | 5.48 | 32.13 |
| Qwen2.5-3B | 4.09 | 6.82 | 2.72 | 4.55 | 8.90 | 33.86 |
| Qwen2.5-7B | 41.55 | 52.62 | 34.20 | 42.79 | 11.99 | 49.23 |
| SmolLM2-135M | 3.68 | 1.30 | 3.17 | 2.72 | 1.42 | 9.37 |
| SmolLM2-360M | 0.03 | 0.03 | 0.06 | 0.04 | 2.52 | 10.91 |
| SmolLM2-1.7B | 0.13 | 0.19 | 0.32 | 0.21 | 4.97 | 15.25 |

the Finance, Medical, and Coding challenges, the Qwen and Mistral models appear to be an optimal choice. Notably, the SmolLM2 model family stands out showing strong model performance relative to their VRAM usage.

Similarly, figures 8 – 11 show the Pareto plots for the different base models (Instruct version) when federated fine-tuned for the different tasks, which are computed from tables 3 – 6. When evaluating performance versus communication costs, the Qwen family of models again stands out for the General NLP and Medical challenges, whereas the Llama model families dominate the frontier with a strong Pass@1 scores relative to the communication costs. Depending on the downstream tasks, the SmolLM model family show a strong performance for the Medical challenge with a fraction of the communication cost. Interestingly, when evaluating performance versus VRAM costs, we observe a similar behaviour where the Qwen model family stands out for the General NLP and Medical challenges. The Mistral-7B-Instruct-v0.3 model is a strong contender, dominating the Finance challenge and lying on the frontier for the remaining challenges. Similar to the non-instruct results, the SmolLM2 model families tend to perform reasonably well for the VRAM requirements.

## B.3   Analysis on 24B base models

Additionally, we extend our investigation to the upper bounds of federated fine-tuning by experimenting with state-of-the-art 24B-parameter models—Mistral-Small-24B-Base-2501 [3] and Mistral-Small-24B-Instruct-2501 [4]—which have been reported to achieve performance comparable to Llama-3.3-70B [4].

Table 13: **Comparison of different non-instruct base models federated fine-tuned on the Finance challenge.** The accuracy values (%) are reported on different downstream tasks. Comm. represents the total communication costs over FL fine-tuning, while Mem. stands for the VRAM costs per client. The highest average performance is indicated in **bold**, while the second-highest is underlined.

| Models | FPB (%) | FIQA (%) | TFNS (%) | Average (%) | Comm. (GB) | Mem. (GB) |
|---|---|---|---|---|---|---|
| Mistral-7B-v0.3 | 73.60 | 68.09 | 39.82 | 60.50 | 30.77 | 17.24 |
| Gemma2-9B | 27.89 | 59.54 | 20.64 | 36.02 | 38.80 | 43.09 |
| Phi-4 (14B) | 29.13 | 61.51 | 26.51 | 39.05 | 126.93 | 40.81 |
| Llama3.2-1B | 46.12 | 42.11 | 40.58 | 42.94 | 8.28 | 12.64 |
| Llama3.2-3B | 59.74 | 22.04 | 60.80 | 47.53 | 17.63 | 16.40 |
| Llama3.1-8B | 60.31 | 39.14 | 61.35 | 53.60 | 30.77 | 27.30 |
| Qwen2-0.5B | 51.65 | 53.95 | 37.77 | 47.79 | 6.54 | 13.94 |
| Qwen2.5-1.5B | 65.18 | 35.53 | 52.05 | 50.92 | 13.69 | 17.54 |
| Qwen2.5-3B | 74.75 | 51.32 | 75.08 | **67.05** | 22.26 | 20.27 |
| Qwen2.5-7B | 71.86 | 52.30 | 74.20 | 66.12 | 29.97 | 28.10 |
| SmolLM2-135M | 29.54 | 57.89 | 22.24 | 36.56 | 3.55 | 5.28 |
| SmolLM2-360M | 50.74 | 24.34 | 51.63 | 42.24 | 6.30 | 6.36 |
| SmolLM2-1.7B | 46.95 | 45.07 | 48.87 | 46.96 | 12.42 | 10.41 |

Table 14: **Comparison of different non-instruct base models federated fine-tuned on the Medical challenge.** The accuracy values (%) are reported on different downstream tasks. Comm. represents the total communication costs over FL fine-tuning, while Mem. stands for the VRAM costs per client. The highest average performance is indicated in **bold**, while the second-highest is underlined.

| Models | PubMedQA (%) | MedMCQA (%) | MedQA (%) | CareQA (%) | Average (%) | Comm. (GB) | Mem. (GB) |
|---|---|---|---|---|---|---|---|
| Mistral-7B-v0.3 | 64.80 | 0.05 | 1.26 | 0.02 | 16.53 | 12.31 | 20.56 |
| Gemma2-9B | 61.00 | 26.89 | 27.18 | 27.22 | **35.57** | 15.52 | 50.57 |
| Phi-4 (14B) | 62.60 | 11.40 | 8.88 | 32.24 | 28.78 | 50.77 | 46.24 |
| Llama3.2-1B | 24.20 | 2.06 | 0.16 | 1.07 | 6.87 | 3.31 | 20.07 |
| Llama3.2-3B | 0.20 | 0.02 | 0.08 | 0.00 | 0.08 | 7.05 | 24.23 |
| Llama3.1-8B | 24.20 | 2.06 | 0.16 | 1.07 | 6.87 | 12.31 | 36.31 |
| Qwen2-0.5B | 51.80 | 10.11 | 13.90 | 9.50 | 21.33 | 2.61 | 18.53 |
| Qwen2.5-1.5B | 0.00 | 8.99 | 2.04 | 6.85 | 4.47 | 5.48 | 21.36 |
| Qwen2.5-3B | 27.80 | 6.26 | 2.28 | 2.54 | 9.72 | 8.90 | 24.79 |
| Qwen2.5-7B | 33.40 | 17.62 | 18.85 | 30.21 | 25.02 | 11.99 | 39.53 |
| SmolLM2-135M | 2.20 | 18.81 | 8.80 | 17.75 | 11.89 | 1.42 | 6.88 |
| SmolLM2-360M | 7.20 | 0.05 | 0.08 | 0.02 | 1.84 | 2.52 | 7.65 |
| SmolLM2-1.7B | 0.80 | 3.87 | 18.38 | 1.23 | 6.07 | 4.97 | 12.04 |

Table 16 presents the evaluation results of both models across the four challenges. As expected, the instruct-tuned model outperforms the non-instruct base model across all tasks. In particular, Mistral-24B-Instruct achieves state-of-the-art performance in the general NLP (65.28%) and coding (63.62%) challenges, while also delivering competitive results in the finance challenge (83.06%). However, its performance on the medical QA task remains relatively modest.

From a system perspective, despite the model's large size, the use of a reduced DoRA configuration—compared to the settings in Table 11—results in manageable communication and memory overhead. This enables efficient training on a single NVIDIA H100 NVL GPU (94 GB).

## B.4   Analysis on domain-specific pre-trained base models

Table 17 presents the evaluation results of several domain-specific pre-trained base models after federated fine-tuning on medical and coding tasks. These models were initially fine-tuned on domain-relevant data (medical or coding) by their respective providers prior to federated training. As shown, they outperform general-purpose base models reported in Tables 3–6, which is expected given their prior exposure to domain-specific knowledge before participating in federated fine-tuning.

Table 15: **Comparison of different non-instruct base models federated fine-tuned on the Coding challenge.** The pass@1 scores (%) are reported on different downstream tasks. Comm. represents the total communication costs over FL fine-tuning, while Mem. stands for the VRAM costs per client. The highest average performance is indicated in **bold**, while the second-highest is underlined.

| Models | MBPP (%) | HumanEval (%) | MultiPL-E (JS) (%) | MultiPL-E (C++) (%) | Average (%) | Comm. (GB) | Mem. (GB) |
|---|---|---|---|---|---|---|---|
| Mistral-7B-v0.3 | 40.20 | 30.49 | 36.02 | 28.57 | 33.82 | 6.15 | 22.27 |
| Gemma2-9B | 50.40 | 49.39 | 44.72 | 39.75 | 46.07 | 7.76 | 56.58 |
| Phi-4 (14B) | 62.40 | 34.76 | 35.40 | 52.80 | **46.34** | 25.39 | 47.04 |
| Llama3.2-1B | 20.40 | 17.68 | 14.29 | 16.15 | 17.13 | 1.66 | 20.89 |
| Llama3.2-3B | 33.40 | 27.44 | 24.84 | 19.25 | 26.23 | 3.53 | 24.99 |
| Llama3.1-8B | 45.60 | 35.37 | 45.34 | 37.89 | 41.05 | 6.15 | 36.96 |
| Qwen2-0.5B | 19.00 | 10.37 | 16.77 | 16.15 | 15.57 | 1.31 | 18.37 |
| Qwen2.5-1.5B | 42.40 | 9.15 | 39.75 | 26.71 | 29.50 | 2.74 | 22.48 |
| Qwen2.5-3B | 40.00 | 15.85 | 42.24 | 36.65 | 33.68 | 4.45 | 27.03 |
| Qwen2.5-7B | 56.60 | 23.17 | 52.17 | 49.07 | 45.25 | 5.99 | 39.36 |
| SmolLM2-135M | 3.40 | 4.27 | 4.97 | 3.11 | 3.94 | 0.71 | 7.01 |
| SmolLM2-360M | 21.00 | 14.02 | 12.42 | 9.94 | 14.35 | 1.26 | 8.24 |
| SmolLM2-1.7B | 31.80 | 23.17 | 24.22 | 23.60 | 25.70 | 2.48 | 11.87 |

Table 16: **Evaluation performance using Mistral-24B models on four challenges.** "SS" refers to Social Sciences. For the medical challenge, "PMQA", "MMCQA", "MQA" and "CQA" correspond to PubMedQA, MedMCQA, MedQA, and CareQA, respectively. For the coding challenge, "HE", "M-JS" and "M-C++" represent HumanEval, MultiPL-E (JS), and MultiPL-E (C++).

| | General NLP | | | | | | Medical | | | | | | |
|---|---|---|---|---|---|---|---|---|---|---|---|---|---|
| Models | STEM | SS | Humanities | Avg | Comm. | Mem. | PMQA | MMCQA | MQA | CQA | Avg | Comm. | Mem. |
| Mistral-24B-Base-2501 | 52.81 | 77.90 | 45.04 | 58.58 | 13.66 | 64.85 | 71.80 | 1.15 | 0.86 | 1.73 | 18.88 | 13.66 | 54.24 |
| Mistral-24B-Instruct-2501 | 54.14 | 79.43 | 62.27 | 65.28 | 13.66 | 65.22 | 72.20 | 26.20 | 30.71 | 44.24 | 43.34 | 13.66 | 54.50 |

| | Finance | | | | | | Code | | | | | | |
|---|---|---|---|---|---|---|---|---|---|---|---|---|---|
| Models | FPB | FIQA | TFNS | Avg | Comm. | Mem. | MBPP | HE | M-JS | M-C++ | Avg | Comm. | Mem. |
| Mistral-24B-Base-2501 | 80.94 | 84.21 | 78.39 | 81.18 | 34.16 | 47.87 | 56.40 | 42.68 | 55.28 | 45.96 | 50.08 | 6.83 | 58.99 |
| Mistral-24B-Instruct-2501 | 85.97 | 80.59 | 82.62 | 83.06 | 34.16 | 48.14 | 62.00 | 69.51 | 62.73 | 60.25 | 63.62 | 6.83 | 56.85 |

Table 17: **Performance evaluation using domain-specific pre-trained base models on medical and coding challenges.** Results are based on submissions to the FlowerTune LLM Leaderboard [2].

| Models | MBPP (%) | HumanEval (%) | MultiPL-E (JS) (%) | MultiPL-E (C++) (%) | Average (%) | Comm. (GB) |
|---|---|---|---|---|---|---|
| Deepseek-Coder-7B-Instruct-v1.5 | 56.80 | 64.63 | 55.90 | 57.76 | 58.77 | 2.9 |
| Qwen2.5-Coder-7B-Instruct | 64.00 | 27.43 | 72.67 | 60.24 | 56.08 | 1.5 |
| | PubMedQA (%) | MedMCQA (%) | MedQA (%) | CareQA (%) | Average (%) | Comm. (GB) |
| Bio-Medical-Llama-3-8B | 70.40 | 60.93 | 65.82 | 55.36 | 63.12 | 1.6 |

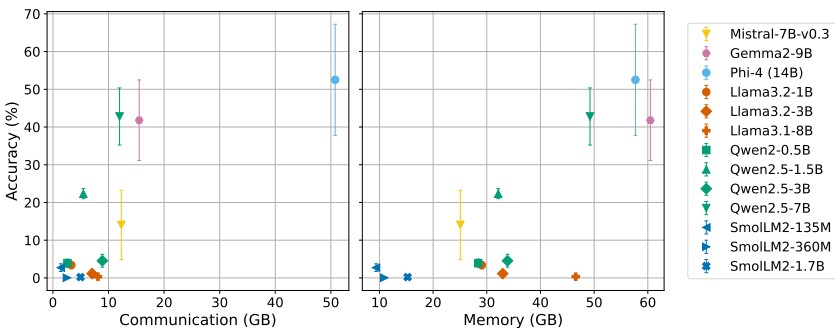

Figure 4: **Accuracy (%) versus system performance for different non-instruct base models federated fine-tuned on the General NLP challenge, presented in table 12.** The errorbar indicates ±1 Std. Dev. on the different downstream tasks.

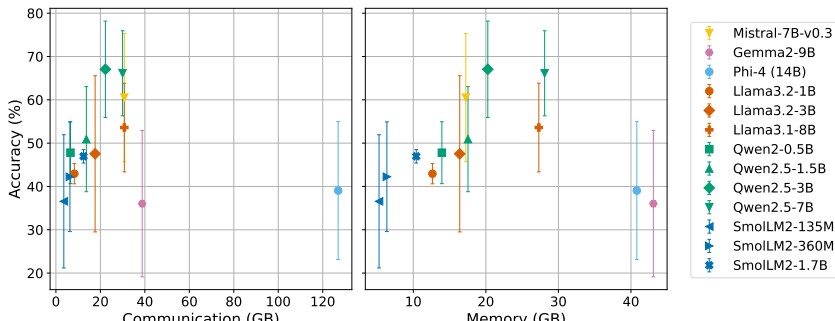

Figure 5: **Accuracy (%) versus system performance for different non-instruct base models federated fine-tuned on the Finance challenge, presented in table 13.** The errorbar indicates ±1 Std. Dev. on the different downstream tasks.

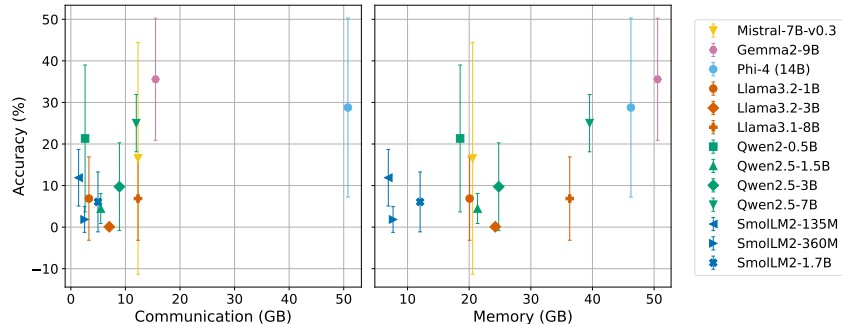

Figure 6: **Accuracy (%) versus system performance for different non-instruct base models federated fine-tuned on the Medical challenge, presented in table 14.** The errorbar indicates ±1 Std. Dev. on the different downstream tasks.

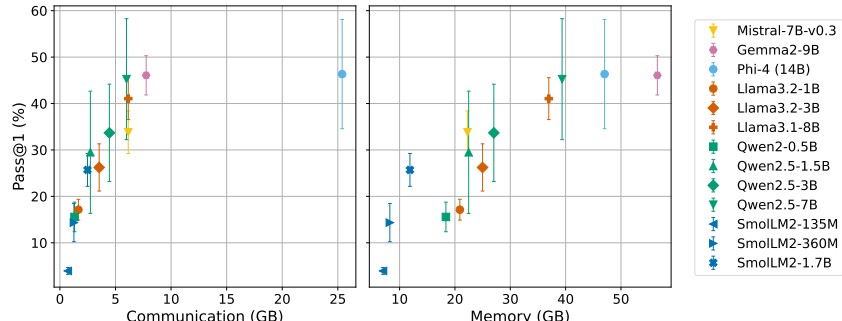

Figure 7: **Pass@1 scores (%) versus system performance for different non-instruct base models federated fine-tuned on the Coding challenge, presented in table 15.** The errorbar indicates ±1 Std. Dev. on the different downstream tasks.

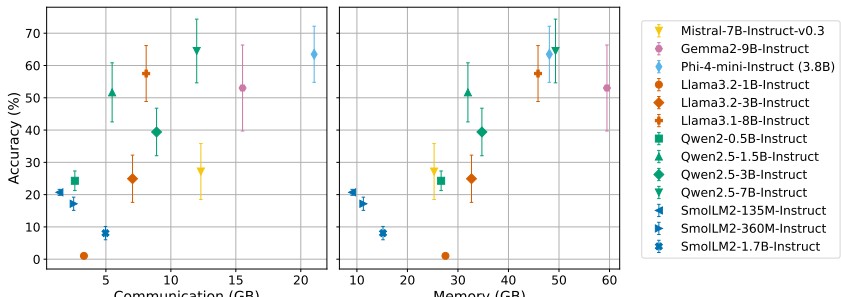

Figure 8: **Accuracy (%) versus system performance for different base models (Instruct version) federated fine-tuned on the General NLP challenge, presented in table 3.** The errorbar indicates ±1 Std. Dev. on the different downstream tasks.

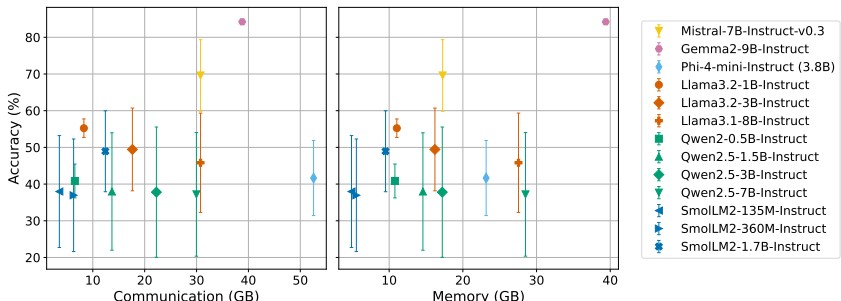

Figure 9: **Accuracy (%) versus system performance for different base models (Instruct version) federated fine-tuned on the Finance challenge, presented in table 4.** The errorbar indicates ±1 Std. Dev. on the different downstream tasks.

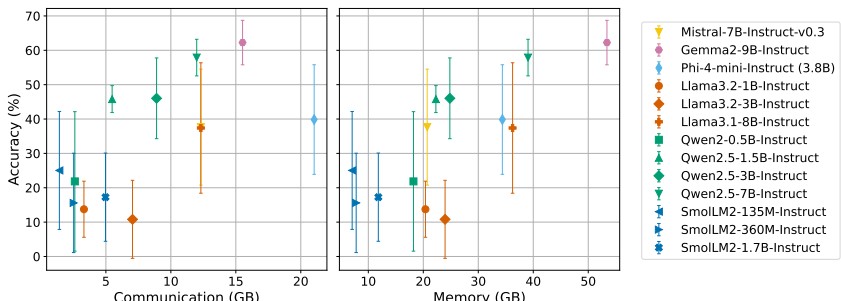

Figure 10: **Accuracy (%) versus system performance for different base models (Instruct version) federated fine-tuned on the Medical challenge, presented in table 5.** The errorbar indicates ±1 Std. Dev. on the different downstream tasks.

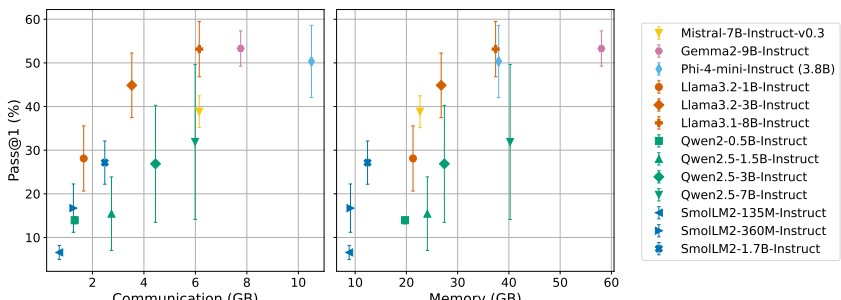

Figure 11: **Pass@1 scores (%) versus system performance for different base models (Instruct version) federated fine-tuned on the Coding challenge, presented in table 6.** The errorbar indicates ±1 Std. Dev. on the different downstream tasks.

# C   Example samples from training datasets

Tables 18–21 show the example samples from the training datasets used in four challenges.

Table 18: Example sample from the training dataset used in **GeneralNLP** challenge.

> *# Instruction*
> What are the three primary colors?
>
> *# Response*
> The three primary colors are red, blue, and yellow. These colors are called primary because they cannot be created by mixing other colors and all other colors can be made by combining them in various proportions. In the additive color system, used for light, the primary colors are red, green, and blue (RGB).

Table 19: Example sample from the training dataset used in **Finance** challenge.

> *# Instruction*
> Google parent Alphabet Inc. reported revenue and earnings that fell short of analysts' expectations, showing the company's search advertising juggernaut was not immune to a slowdown in the digital ad market. The shares fell more than 6%.
>
> *# Response*
> Moderately negative

Table 20: Example sample from the training dataset used in **Medical** challenge.

> *# Instruction*
> What is the relationship between very low Mg2+ levels, PTH levels, and Ca2+ levels?
>
> *# Response*
> Very low Mg2+ levels correspond to low PTH levels which in turn results in low Ca2+ levels.

Table 21: Example sample from the training dataset used in **Coding** challenge.

> *# Instruction*
> Write a replace method for a string class which replaces the given string with a given set of characters. string = "Hello World!" replace_with = "Greetings!"
>
> *# Response*
> ```
> def replace(self, replace_with):
> new_string = ""
> for char in self:
> if char == " ":
> new_string += replace_with
> else:
> new_string += char
> return new_string
> ```

# D  How to participate in the FlowerTune LLM Leaderboard?

The FlowerTune LLM Leaderboard is open for ongoing submissions. Please refer to Figure 12 for a guide on how to participate. Detailed instructions and submission guidelines are available on the official leaderboard website [2]. In addition to contributions to the existing challenges, we welcome proposals for new settings and tasks. We look forward to your participation and collaboration in advancing the frontiers of federated LLM fine-tuning.

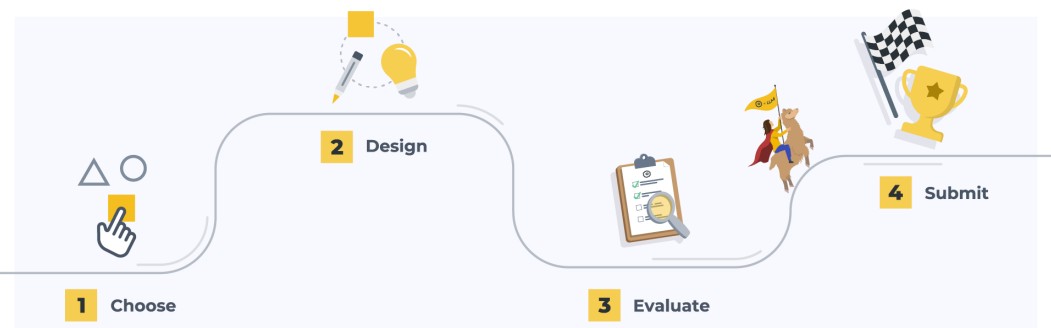

Figure 12: Workflow for participating in the FlowerTune LLM Leaderboard. (1) Explore the available challenge boards and select a challenge; (2) propose and implement a federated fine-tuning approach using the provided template code; (3) evaluate the performance of your fine-tuned LLM; (4) submit your results to claim a position on the leaderboard.

## D.1  Overview of GPUs used by participants

Submissions to the FlowerTune LLM Leaderboard have come from a number of institutions and individual contributors, and Figure 13 shows which GPUs were used more frequently. The NVIDIA A100 (40GB) and A40 (which has 48GB of VRAM) were the most popular cards. This is unsurprising as that amount of VRAM is needed to fine-tune the largest LLMs considered in these leaderboards. These cards are also widely available in all major cloud providers.

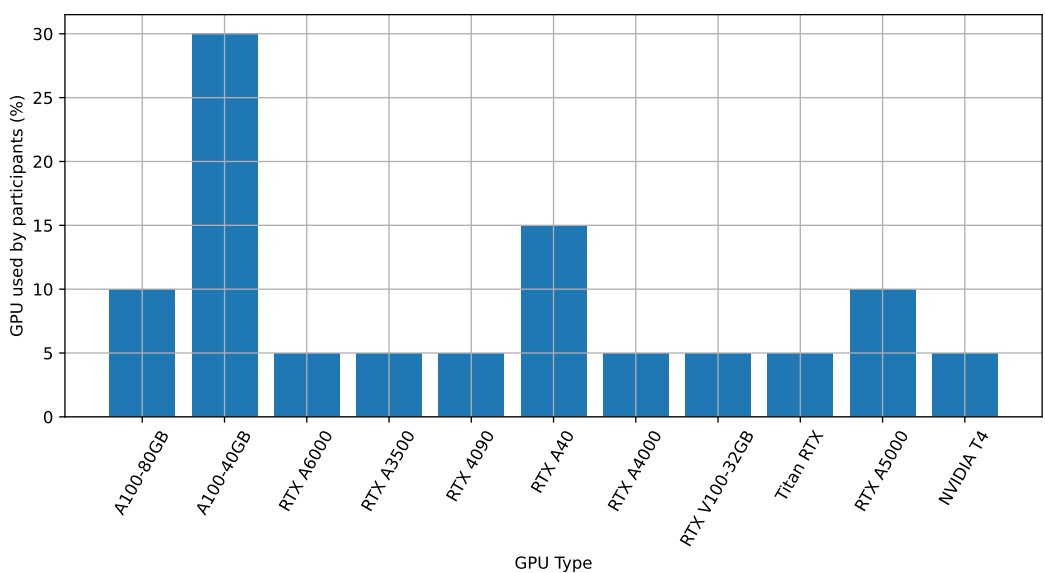

Figure 13: GPU types used across submissions from institutional and individual community contributors.

