# OpenReview forum: "FlowerTune: A Cross-Domain Benchmark for Federated Fine-Tuning of Large Language Models"
_NeurIPS.cc/2025/Datasets_and_Benchmarks_Track — NeurIPS 2025 Datasets and Benchmarks Track poster_

### Official Review · Reviewer_YKsT · 2025-06-26

**Rating:** 5
**Confidence:** 5

**Summary:**

This paper introduces FlowerTune, a pioneering benchmark suite for evaluating the federated fine-tuning of Large Language Models (LLMs). Amid growing demand for domain-specialized and privacy-preserving AI, Federated Learning (FL) has emerged as a highly promising solution. However, the performance and suitability of existing pre-trained LLMs in FL settings have been largely unexplored. To fill this gap, the authors have constructed a comprehensive benchmark spanning four key domains: general NLP, finance, medical, and code. Each domain is equipped with a dedicated federated instruction-tuning dataset and downstream evaluation tasks. Through an open-source and community-driven approach, this work presents the first large-scale comparative study of 26 pre-trained LLMs of varying sizes, combining multiple federated aggregation and parameter-efficient fine-tuning strategies (e.g., DoRA/LoRA). This work not only provides the community with a reproducible and standardized evaluation platform but also offers invaluable practical insights into model performance, resource constraints, and domain adaptation, laying a solid foundation for the development of real-world, privacy-preserving, domain-specialized LLMs.

**Dataset Code Accessibility:**

Yes

**Ethical Considerations:**

No, there are no or only very minor ethics concerns

**Limitations Weaknesses:**

* **Uniformity of Hyper-parameter Tuning:** To ensure a fair comparison, all experiments were conducted under a unified FL configuration and hyper-parameter set. While necessary for establishing a baseline, this means that the optimal hyper-parameters for each specific model-dataset pair may not have been found. Future work could build upon this benchmark to explore adaptive or model-specific hyper-parameter tuning strategies, which might unlock further performance gains for certain models.
* **Performance of Aggregation Strategies:** The experimental results indicate that several classic federated aggregation algorithms (FedAvg, FedProx, etc.) show only marginal performance differences in the context of LLM adapter tuning. This is not a flaw of the paper but rather a valuable finding. It suggests a need for the community to develop more targeted aggregation algorithms tailored to this specific parameter space (low-rank adapters) to more effectively fuse knowledge from diverse clients, which is a promising research direction.

**Strengths Contributions:**

* **Vision and Importance:** This work astutely identifies a critical bottleneck in LLM development: the reliance on massive public datasets and the privacy challenges associated with sensitive domains like finance and medicine. By introducing FlowerTune, the first public benchmark dedicated to federated LLM fine-tuning, this work significantly advances the field of privacy-preserving AI. This contribution is groundbreaking, offering the academic and industrial communities a much-needed, standardized testbed to explore and validate various federated fine-tuning strategies.
* **Comprehensive Experiments and Deep Insights:** The core strength of this work lies in its exceptionally thorough and systematic experimental evaluation. The authors benchmarked an extensive set of 26 pre-trained models (both base and instruction-tuned versions) with parameter sizes ranging from 135M to 14B. The experiments cover four heterogeneous domains and systematically compare various federated aggregation algorithms (e.g., FedAvg, FedProx) and parameter-efficient fine-tuning (PEFT) techniques (LoRA, DoRA). The resulting analysis is profound and offers practical guidance, for instance:
  * The experiments clearly demonstrate that instruction-tuned models (-Instruct versions) generally outperform their non-instructed counterparts in federated settings, showing more stable and superior performance.
  * The study reveals that while larger models typically perform better, some small to medium-sized models (e.g., `LLaMA-3.2-3B`) can achieve performance comparable to that of larger models (e.g., `LLaMA-3.1-8B`) on specific tasks like financial classification. This is a crucial finding for resource-constrained FL applications. For example, on the general NLP task, `Qwen2.5-7B-Instruct` achieved an average accuracy of 64.50%, surpassing even the larger `Gemma2-9B-Instruct` (53.03%).
* **Pragmatic Evaluation of System Performance:** Beyond model accuracy, this work meticulously evaluates key system-level metrics during federated fine-tuning, namely communication cost and client-side VRAM usage. By employing PEFT methods like DoRA/LoRA, the study shows that federated fine-tuning of even large models (up to 14B, and even 24B) is feasible within the memory capacity of a single modern GPU (e.g., A100 80GB). For example, `SmolLM2-135M-Instruct` achieves acceptable performance while requiring only about 9.06 GB of VRAM. This provides critical feasibility evidence and performance-resource trade-off analysis for deploying federated LLM fine-tuning on real-world edge devices.
* **High-Quality Open-Source Contribution:** The authors have delivered more than just a leaderboard; they have provided a complete, open-source, and reproducible research framework. This includes partitioned federated datasets for four domains, end-to-end training and evaluation code, and an extensible template that allows community members to easily integrate new models and algorithms. This open and collaborative approach has significantly lowered the barrier to entry in this field and has already attracted numerous community contributions, making its value extend far beyond the paper itself.

---

> ### Author Rebuttal · Authors · 2025-07-31
>
> We sincerely thank you for your thoughtful comments and for recognizing the novelty and importance of our study. We have addressed all of your concerns in detail below.
>
> ## Regarding the uniformity of hyper-parameter tuning
> Thank you for highlighting this point. In this study, our primary goal is to provide a standardized testbed that allows researchers and developers to evaluate and compare fine-tuned models under consistent conditions. To that end, the hyperparameters used in our experiments (e.g. Table 3-6) are a judicious selection based on those commonly employed in leaderboard submissions, ensuring broad applicability across different models and domains.
>
> We fully acknowledge that optimizing hyperparameters for each specific model–dataset pair is valuable. In fact, many contributors to the leaderboard are actively pursuing such optimizations (please checkout the current submissions in the leaderboard for reference). We hope this can serve as a solid foundation for further exploration and fine-tuning by the community. In future iterations of the benchmark, we plan to investigate more tailored hyperparameter configurations that are specific to particular model–domain combinations.
>
> ## Regarding the performance of aggregation strategies
>
> Thank you for pointing this out. In the updated version, we will expand on the observations derived from this study and explicitly include this point in the future work section to raise awareness of this limitation within the broader research community.

---

> > ### Comment · Reviewer_YKsT · 2025-08-04
> >
> > Thanks for the rebuttal, I will keep the score.

---

### Official Review · Reviewer_zXAJ · 2025-07-01

**Rating:** 4
**Confidence:** 3

**Summary:**

The paper introduces FlowerTune, the first cross-domain benchmark for federated learning (FL) of Large Language Models (LLMs). FlowerTune addresses challenges like data privacy domain specialization by enabling decentralized model training. The paper evaluated 26 pre-trained LLMs across four sensitive domains,  and found that larger models (e.g., Gemma2-9B) generally outperform smaller ones but incur higher resource costs, while smaller models excel in resource-constrained FL settings for simpler tasks.

**Additional Feedback:**

Please improve the paper by addressing the concerns in the weakness.

**Dataset Code Accessibility:**

Yes

**Dataset Code Comments:**

The paper provides an anonymous link to the code and dataset downloading scripts.

**Ethical Comments:**

The paper studies a benchmark for LLM tuning in FL, and the datasets are from public dataset.

**Ethical Considerations:**

No, there are no or only very minor ethics concerns

**Final Justification:**

I have read the authors' rebuttal and the rebuttal has addressed my concerns.

**Limitations Weaknesses:**

**Evaluation**: The evaluation is mainly based on static hyperparameters. For example, the paper adopts uniform training settings for all models (Table 10), potentially obscuring model-specific optima. With tuned leraning rate, the Qwen2.5-7B’s subpar finance accuracy (37.20% in Table 4) may improve.

**Narrow Domain Coverage**: The paper only include four important domains but does not explain why other domains are not considered.  I suggest the authors to explain why not include other domains and the relation between the selected domains and other domains.

**Strengths Contributions:**

**Significance & Novelty**: The paper proposes the first cross-domain FL benchmark for LLM fine-tuning (Sec 1-2).

**Novel contribution**: The paper introduces benchamrk dataset across four important domains with tailored metrics, alllowing the future researcher to conduct standardized comparison missing in prior FL-LLM work.

**Potential Impact**:
The paper includes an open-source leaderboard to promote reproducibility, enabling the community to better assess LLM tuning under FL.

---

> ### Author Rebuttal · Authors · 2025-07-31
>
> We sincerely thank you for your thoughtful comments and for recognizing the novelty and importance of our study. We have addressed all of your concerns in detail below.
>
> ## Regarding the concern of static hyperparameters
>
> Thank you for highlighting this point. In this study, our primary goal is to provide a standardized testbed that allows researchers and developers to evaluate and compare fine-tuned models under consistent conditions. To that end, the hyperparameters used in our experiments (e.g. Table 3-6) are a judicious selection based on those commonly employed in leaderboard submissions, ensuring broad applicability across different models and domains.
>
> We fully acknowledge that optimizing hyperparameters for each specific model–dataset pair is valuable. In fact, many contributors to the leaderboard are actively pursuing such optimizations (please checkout the current submissions in the leaderboard for reference). We hope this can serve as a solid foundation for further exploration and fine-tuning by the community. In future iterations of the benchmark, we plan to investigate more tailored hyperparameter configurations that are specific to particular model–domain combinations.
>
> ## Regarding the concern of narrow domain coverage
>
> Thank you for raising this point. The four selected domains (general NLP, finance, medical, and code) are particularly important because they involve multiple institutions (e.g., medical, financial, and educational organizations) collaborating to train models without sharing sensitive or proprietary data. These fields often handle highly confidential information (e.g., patient records, financial transactions, internal documents, or source code) that cannot be centralized due to strict privacy regulations (e.g., GDPR, HIPAA), competitive concerns, and data ownership constraints.
>
> Additionally, these domains are widely regarded as standard benchmarks for evaluating the performance of different LLMs and have been adopted in many prior studies [1, 2, 3].
>
> We selected these four domains as the initial set of challenges for this benchmark. However, the leaderboard is ongoing, and we plan to include additional domains in future releases.
>
> [1] Ye, Rui, et al. "Openfedllm: Training large language models on decentralized private data via federated learning." *Proceedings of the 30th ACM SIGKDD conference on knowledge discovery and data mining*. 2024.
>
> [2] Wang, Yidong, et al. "PandaLM: An Automatic Evaluation Benchmark for LLM Instruction Tuning Optimization." *ICLR*. 2024.
>
> [3] Tan, Sijun, et al. "JudgeBench: A Benchmark for Evaluating LLM-Based Judges." *The Thirteenth International Conference on Learning Representations*.

---

> ### Author Response · Authors · 2025-08-06
>
> Thank the reviewer again for taking the time to review our work. If the reviewer has any further questions or concerns, we would be more than happy to discuss them in detail.

---

> > ### Author Response · Authors · 2025-08-08
> >
> > Thank you again for your thoughtful comments. If you have any additional questions or concerns, we would be glad to address them in detail. Otherwise, we would greatly appreciate your consideration in updating your score.

---

### Official Review · Reviewer_jo8b · 2025-07-01

**Rating:** 5
**Confidence:** 3

**Summary:**

This paper presents a significant contribution to the field of federated learning for large language models (LLMs) by introducing FlowerTune, a comprehensive cross-domain benchmark for evaluating federated fine-tuning.

**Dataset Code Accessibility:**

Yes

**Ethical Comments:**

There are no ethical concerns.

**Ethical Considerations:**

No, there are no or only very minor ethics concerns

**Final Justification:**

I keep the positive for this paper.

**Limitations Weaknesses:**

1) Discuss how VRAM/communication constraints (e.g., 80 GB VRAM limit) impact deployment on edge devices. Provide estimates for more resource-limited settings.
2) Fix inconsistent model naming (e.g., "Llama-3.1-8B" vs. "LLaMA-3.1-8B" in Tables 3–6).
3) Provide standard deviations for key results (Tables 3–6) to assess statistical significance.

**Strengths Contributions:**

1) First benchmark for FL fine-tuning of LLMs across diverse domains (general NLP, finance, medical, coding).
2) Large-scale evaluation of 26 base models under unified FL settings, offering actionable insights for model selection, resource constraints, and domain adaptation.
3) Well-designed pipelines for federated instruction tuning and domain-specific evaluation (e.g., MMLU for NLP, PubMedQA for medical).
4) Open-source, community-driven platform (FlowerTune LLM Leaderboard) with standardized tools, datasets, and baselines for reproducibility.

---

> ### Author Rebuttal · Authors · 2025-07-31
>
> We sincerely thank you for your thoughtful comments and for recognizing the novelty and importance of our study. We have addressed all of your concerns in detail below.
>
> ## Regarding the deployment on edge devices and estimation
>
> Thank you for raising this point. In this paper, we focus on cross-silo federated settings, as the current training pipeline for LLMs still requires significant GPU resources, making training on edge devices challenging. We provide memory cost estimates for different LLMs, which suggest that it may be feasible to train smaller models (e.g., SmolLM2, Qwen2-0.5B, Llama3.2-1B) on edge devices such as an NVIDIA Jetson board, MacBook, or Raspberry Pi with 16 GB of RAM—when considering memory consumption alone. However, compute speed is another critical factor. This can only be accurately measured on real edge devices and is typically much slower than GPU-based training. We will add this analysis and discussion to the camera-ready version of the paper.
>
> ## Regarding inconsistent model naming
>
> Thank you for pointing it out. We will fix model naming throughout the paper for the camera-ready version.
>
> ## Regarding the assessment of statistical significance
>
> Thank you for raising this important point. At the time of our experiments, we opted not to perform multiple runs for all models due to the substantial computational cost, and chose a set of judicious hyperparameters to run our benchmarks. Additionally, federated fine-tuning is inherently more time-consuming than centralized training, as it involves extra steps for model communication and aggregation. This decision is consistent with the approach taken in prior NeurIPS benchmark papers [1, 2, 3].
>
> To address the sensitivity of model performance, we evaluated each model on multiple datasets within each domain. This allows us to assess robustness across a diverse range of evaluation scenarios. We report both average performance and provide error bars in Figures 5–11 to illustrate variability across datasets. We will highlight this point in the camera-ready version of the paper.
>
> [1] Mogrovejo, David Orlando Romero, et al. "CVQA: Culturally-diverse Multilingual Visual Question Answering Benchmark." *The Thirty-eight Conference on Neural Information Processing Systems Datasets and Benchmarks Track*.
>
> [2] Tian, Minyang, et al. "Scicode: A research coding benchmark curated by scientists." *Advances in Neural Information Processing Systems* 37 (2024): 30624-30650.
>
> [3] Ye, Rui, et al. "Fedllm-bench: Realistic benchmarks for federated learning of large language models." *Advances in Neural Information Processing Systems* 37 (2024): 111106-111130.

---

> ### Author Response · Authors · 2025-08-06
>
> Thank the reviewer again for taking the time to review our work. If the reviewer has any further questions or concerns, we would be more than happy to discuss them in detail.

---

### Official Review · Reviewer_ojVP · 2025-07-03

**Rating:** 5
**Confidence:** 4

**Summary:**

This paper introduces the "FlowerTune LLM Leaderboard," a novel benchmark designed to systematically evaluate the performance of Large Language Models (LLMs) when fine-tuned in a Federated Learning (FL) setting. The work addresses critical challenges in LLM development, e.g. data privacy concerns associated with accessing domain-specific, sensitive information. The benchmark spans four diverse and high-impact application domains: general Natural Language Processing (NLP), finance, medical, and coding. For each domain, it provides federated instruction-tuning datasets and tailored domain-specific evaluation metrics. The authors present a comprehensive comparison of 26 pre-trained LLMs, exploring various aggregation algorithms and fine-tuning strategies within the federated paradigm. The results offer insights into model performance, resource constraints (such as communication overhead and VRAM usage), and domain adaptation capabilities. This foundational work aims to accelerate the development of privacy-preserving, domain-specialized LLMs for real-world applications.

**Additional Feedback:**

Extra questions:

1. Fine-tuning methods only include Lora and Dora, support QloRA or other efficient fine-tuning methods will be better, because efficiency is an important factor.

2. The paper states that the datasets were partitioned to "emulate data distributions that could be expected across institutions in FL environments". Could the authors elaborate on whether a quantitative analysis of the data heterogeneity within these simulated federated datasets was performed? Specifically, were common non-IID characteristics such as label imbalance, feature distribution shifts, or data volume disparities across clients assessed? If so, could you provide specific metrics or data points to describe the degree of this heterogeneity for each domain? This would be invaluable for interpreting the generalizability of model and strategy performance to complex,

**Dataset Code Accessibility:**

Yes

**Dataset Code Comments:**

The code and benchmark is available.

**Ethical Considerations:**

No, there are no or only very minor ethics concerns

**Final Justification:**

The author's rebuttal has solved my concerns.  I recommand the positive for this paper.

**Limitations Weaknesses:**

1. This paper lacks a detailed quantitative characterization of the actual data heterogeneity (e.g., non-IID properties, class imbalance, domain shift metrics) within the federated settings. Without explicitly quantifying the degree of data heterogeneity across clients for each domain, it becomes challenging to fully interpret how different models and FL strategies perform under varying levels of real-world data non-IIDness, which is a fundamental challenge in federated learning.

2. The authors state that "All experiments are conducted using a unified configuration without task-specific or model-specific hyper-parameter tuning". While this ensures fairness, the repeated observation of numerical differences in performance tables (e.g., Tables 3-6) and the reliance on single runs (implied by the fixed seed for reproducibility as stated in the checklist) raise concerns about the statistical significance of these differences. Although error bars are presented in the Appendix figures, their absence from the main results tables makes it difficult for readers to gauge the reliability of direct numerical comparisons. For a benchmark, demonstrating statistically significant differences, or acknowledging their absence, is crucial for drawing robust conclusions.

3. Relative Limited Scope of Federated Learning Scenarios: The study primarily focuses on "cross-silo federated settings". While this is a common and important FL paradigm, a comprehensive benchmark could benefit from exploring a broader spectrum of FL scenarios, such as cross-device settings with more extreme resource constraints (e.g., lower computation, more volatile connectivity, or heterogeneous hardware beyond GPUs mentioned in Appendix D.1 ) and potentially more complex client participation dynamics. The current focus, while valuable, may not fully capture the challenges and optimal solutions for all real-world federated LLM deployment contexts.

**Strengths Contributions:**

1. A Critical Research : The paper effectively targets a highly relevant area at the intersection of LLMs and Federated Learning. As LLMs continue to scale and data privacy becomes paramount, the ability to fine-tune these models on decentralized, sensitive data without compromising privacy is crucial. This work pioneers a systematic evaluation in this under-explored domain, offering a much-needed standardized framework.

2. Comprehensive Benchmark Design: Cross-Domain Applicability: The inclusion of four distinct and high-impact domains (general NLP, finance, medical, and coding) is a significant strength, demonstrating the benchmark's versatility and relevance to diverse real-world applications involving sensitive data. This broad coverage provides a more holistic understanding of FL LLM performance than single-domain studies.

3. Extensive Model Coverage: Benchmarking 26 diverse pre-trained LLMs, ranging in size from 135 million to 14 billion parameters, offers a diverse scale of empirical comparison. This allows for robust insights into model suitability under FL constraints. Promotion of Open Science and Community Engagement: The establishment of the "FlowerTune LLM Leaderboard" as an open-source platform, with a commitment to continuous updates and maintenance, is highly commendable. This initiative fosters reproducibility, encourages contributions from the broader research community, and creates a living resource that can evolve with the field.

4. The detailed experimental analysis leads to several significant findings that are valuable for practitioners. The identification of models suitable for resource-constrained environments also provides direct guidance for deployment decisions

---

> ### Author Rebuttal · Authors · 2025-07-31
>
> We sincerely thank you for your thoughtful comments and for recognizing the novelty and importance of our study. We have addressed all of your concerns in detail below.
>
> ## Regarding the uniformity of hyper-parameter tuning
>
> Thank you for highlighting this point. In this study, our primary goal is to provide a standardized testbed that allows researchers and developers to evaluate and compare fine-tuned models under consistent conditions. To that end, the hyperparameters used in our experiments (e.g. Table 3-6) are a judicious selection based on those commonly employed in leaderboard submissions, ensuring broad applicability across different models and domains.
>
> We fully acknowledge that optimizing hyperparameters for each specific model–dataset pair is valuable. In fact, many contributors to the leaderboard are actively pursuing such optimizations (please checkout the current submissions in the leaderboard for reference). We hope this can serve as a solid foundation for further exploration and fine-tuning by the community. In future iterations of the benchmark, we plan to investigate more tailored hyperparameter configurations that are specific to particular model–domain combinations.
>
> ## Regarding the assessment of statistical significance
>
> Thank you for raising this important point. At the time of our experiments, we opted not to perform multiple runs for all models due to the substantial computational cost. Additionally, federated fine-tuning is inherently more time-consuming than centralized training, as it involves extra steps for model communication and aggregation. This decision is consistent with the approach taken in prior NeurIPS benchmark papers [1, 2, 3].
>
> To address the sensitivity of model performance, we evaluated each model on multiple datasets within each domain. This allows us to assess robustness across a diverse range of evaluation scenarios. We report both average performance and provide error bars in Figures 5–11 to illustrate variability across datasets. We will highlight this point in the camera ready version of the paper.
>
> [1] Mogrovejo, David Orlando Romero, et al. "CVQA: Culturally-diverse Multilingual Visual Question Answering Benchmark." *The Thirty-eight Conference on Neural Information Processing Systems Datasets and Benchmarks Track*.
>
> [2] Tian, Minyang, et al. "Scicode: A research coding benchmark curated by scientists." *Advances in Neural Information Processing Systems* 37 (2024): 30624-30650.
>
> [3] Ye, Rui, et al. "Fedllm-bench: Realistic benchmarks for federated learning of large language models." *Advances in Neural Information Processing Systems* 37 (2024): 111106-111130.
>
> ## Regarding the quantitative analysis of the data distribution
>
> Thank you for raising this point. In this paper, our FL setup simulates cross-institutional scenarios—such as those in medical, financial, and educational domains—where the number of clients is less compared to cross-device settings. A statistical summary of the fine-tuning dataset splits is provided in Table 1. In addition, we conducted a further quantitative analysis of the client-level data distribution within each domain, detailed below:
>
> **General NLP**
>
> \# of Samples: [2601, 2601, 2600, 2600, 2600, 2600, 2600, 2600, 2600, 2600, 2600, 2600, 2600, 2600, 2600, 2600, 2600, 2600, 2600, 2600]
>
> Instruction Avg Length: [59, 59, 59, 60, 59, 59, 59, 59, 60, 59, 59, 60, 59, 59, 60, 58, 59, 60, 59, 59]
>
> Response Avg Length: [668, 668, 663, 671, 680, 673, 676, 664, 686, 672, 688, 708, 646, 692, 688, 680, 689, 693, 671, 660]
>
> Semantic Similarity: [-0.017059, 0.043205, 0.031212, -0.125241, -0.072221, 0.031201, 0.041594, -0.140428, -0.001119, -0.002795, -0.10299, -0.139953, 0.03993, -0.052652, -0.044695, -0.138351, -0.079399, -0.140759, -0.049251, 0.041702]
>
> **Finance**
>
> \# of Samples: [1536, 1536, 1536, 1536, 1536, 1536, 1536, 1536, 1536, 1536, 1536, 1536, 1536, 1536, 1536, 1536, 1536, 1536, 1536, 1536, 1536, 1536, 1535, 1535, 1535, 1535, 1535, 1535, 1535, 1535, 1535, 1535, 1535, 1535, 1535, 1535, 1535, 1535, 1535, 1535, 1535, 1535, 1535, 1535, 1535, 1535, 1535, 1535, 1535, 1535]
>
> Instruction Avg Length: [224, 225, 223, 223, 224, 223, 225, 223, 220, 226, 226, 224, 223, 226, 228, 225, 223, 226, 223, 227, 225, 225, 228, 225, 221, 225, 223, 225, 223, 221, 224, 224, 225, 226, 223, 226, 224, 226, 221, 226, 227, 223, 223, 220, 228, 229, 224, 227, 226, 224]
>
> Response Avg Length: [9, 9, 9, 9, 9, 9, 9, 9, 9, 9, 9, 9, 9, 9, 9, 9, 9, 9, 9, 9, 9, 9, 9, 9, 9, 9, 9, 9, 9, 9, 9, 9, 9, 9, 9, 9, 9, 9, 9, 9, 9, 9, 9, 9, 9, 9, 9, 9, 9, 9]
>
> Semantic Similarity: [0.715111, 0.469214, 0.285945, 0.647425, 0.514729, 0.790595, 0.792727, 0.352473, 0.330893, 0.671933, 0.778249, 0.141519, 0.790175, 0.782284, 0.763719, 0.70235, 0.697406, 0.767668, 0.712049, 0.125783, 0.783797, 0.578822, 0.717162, 0.791174, 0.639695, 0.785276, 0.646192, 0.629257, 0.458752, 0.310268, 0.792542, 0.609435, 0.689518, 0.7926, 0.790289, 0.785883, 0.769438, 0.786414, 0.240132, 0.406068, 0.768684, 0.33576, 0.771714, 0.256589, 0.736102, 0.761169, 0.792017, 0.5755, 0.771863, 0.774038]
>
> **Medical**
>
> \# of Samples: 1698, 1698, 1698, 1698, 1698, 1698, 1698, 1698, 1698, 1698, 1698, 1698, 1698, 1698, 1698, 1697, 1697, 1697, 1697, 1697
>
> Instruction Avg Length: [92, 92, 92, 90, 92, 93, 94, 91, 92, 91, 92, 91, 91, 92, 92, 92, 91, 93, 92, 92]
>
> Response Avg Length: [346, 361, 346, 331, 340, 357, 362, 343, 343, 343, 353, 354, 355, 345, 351, 358, 341, 346, 344, 352]
>
> Semantic Similarity: [-0.084049, -0.038203, -0.015703, -0.084682, -0.049039, -0.028164, -0.031425, -0.030609, -0.051584, -0.071579, -0.042079, -0.067059, -0.075075, -0.029978, -0.021609, -0.061427, -0.057663, -0.089613, -0.015855, -0.081245]
>
> **Code**
>
> \# of Samples: [2003, 2003, 2002, 2002, 2002, 2002, 2002, 2002, 2002, 2002]
>
> Instruction Avg Length: [96, 96, 101, 97, 97, 99, 97, 98, 98, 98]
>
> Response Avg Length: [199, 191, 193, 202, 193, 201, 200, 190, 196, 200]
>
> Semantic Similarity: [0.034132, -0.097949, -0.241688, -0.26715, -0.052489, -0.148023, 0.052794, 0.028362, 0.037034, -0.138201]
>
> **Explanation of the data:** “# of Samples” indicates the number of data points per client. “Instruction Avg Length” and “Response Avg Length” represent the average sequence lengths of instructions and responses, respectively, for each client. For “Semantic Similarity,” we compute embeddings for each instruction–response pair using the pre-trained all-MiniLM-L6-v2 model and then average these embeddings for each client. After applying dimensionality reduction, we calculate the cosine similarity between each client and all other clients, reporting the average similarity value per client to illustrate semantic variation (with values closer to 1 indicating greater similarity).
>
> From this analysis, we observe that the number of samples is relatively balanced across clients within each domain—a realistic assumption for cross-institutional FL settings. Variation in instruction and response lengths arises primarily from the inherent properties of the source datasets. We did not partition client data based on class distributions, as the instruction-tuning datasets used in this study are unlabeled. However, the semantic similarity metrics reveal meaningful differences across clients in each domain, particularly in the general NLP, medical, and code domains, where similarity scores approach 0, indicating a notable degree of heterogeneity. The higher similarity values observed in the finance domain are likely inherent to the original dataset prior to splitting. We will include this analysis in the camera-ready version of the paper.
>
> We also acknowledge, as noted in the Limitations section, that our current setup does not fully capture the level of heterogeneity seen in more challenging FL scenarios. As future work, we plan to explore a broader range of settings, including more domains with greater distributional shifts.
>
> ## Regarding the concern of “Relative Limited Scope of Federated Learning Scenarios”
>
> In this paper, we focus on cross-silo federated settings because it enables multiple institutions to collaboratively train models without sharing sensitive or proprietary data—particularly relevant in domains such as healthcare, finance, and education. These fields often handle highly confidential information (e.g., patient records, financial transactions, internal documents, or source code) that cannot be centralized due to strict privacy regulations (e.g., GDPR, HIPAA), competitive concerns, and data ownership constraints.
>
> Additionally, the focus of this work is LLM fine-tuning, which currently requires significant GPU resources. Training LLMs on resource-constrained devices, as typically found in cross-device settings, remains highly challenging. Combined with the lower client participation rates inherent in cross-device FL, achieving acceptable model performance in such settings is considerably more challenging.
>
> We acknowledge that exploring scenarios with more extreme resource constraints would be valuable; however, this is beyond the scope of this study. Future work will aim to extend our benchmark to more resource-limited environments.
>
> ## Regarding the experiments for QLoRA
>
> The experiments we conducted are already under QLoRA style (see the implementation of QLoRA — HuggingFace PEFT Quantization documentation). In default, we use 4-bit quantization for the main experiments. We will update the manuscript to clarify this point.

---

> > ### Comment · Reviewer_ojVP · 2025-08-05
> > **Need add more related works discussion.**
> >
> > Thanks for author's rebuttal. The extensive rebuttal has partialy solved most of my concerns. I think authors need to discuss more related-works on On-Device Fine-tuning scenes, due to its aligend objective with Federated Fine-Tuning learning. Add the discussion about those papers[1][2] will have better relate-work review.
> >
> > [1] GSQ-Tuning: Group-Shared Exponents Integer in Fully Quantized Training for LLMs On-Device Fine-tuning, ACL 2025.
> > [2] QEFT: Quantization for Efficient Fine-Tuning of LLMs, ACL 2024.

---

> > > ### Author Response · Authors · 2025-08-05
> > >
> > > Thank you for the suggestion. We will include this discussion in the Related Work section of the updated version of the paper.

---

> > > > ### Comment · Reviewer_ojVP · 2025-08-06
> > > >
> > > > It might be a better approach to engage in substantive discussion during the rebuttal phase rather than offering mere verbal promises. No reviewer is willing to see only verbal assurances without concrete elaboration.

---

> > > > > ### Author Response · Authors · 2025-08-06
> > > > > **Regarding the discussion of on-device fine-tuning**
> > > > >
> > > > > We thank the reviewer for their comment and agree that expanding the discussion on on-device fine-tuning is instructive to readers to provide some contextual comparison to cross-silo fine tuning. To emphasize, as mentioned in the manuscript, our paper focuses on cross-silo federated fine-tuning of LLMs, where participants typically have access to GPU resources. Unlike cross-device settings—where devices are highly resource-constrained and on-device fine-tuning techniques are essential—our scenario assumes more capable compute environments. Nonetheless, we acknowledge that advances in on-device fine-tuning are promising and applicable to cross-silo fine tuning.
> > > > >
> > > > > Indeed, the reviewer's suggested references has surfaced recent studies that explore efficient on-device fine-tuning of LLMs to address privacy and resource constraints. GSQ-Tuning [1] enables fully integer-based on-device LLM fine-tuning using group-shared exponents, while QEFT [2] improves efficiency by updating only sensitive weight columns in mixed precision. These methods extend PEFT methods like LoRA [3] and QLoRA [4] for resource-constrained settings.
> > > > >
> > > > > We have revised our manuscript in the Related Work section as follows:
> > > > > ```
> > > > > Recent work has explored efficient on-device fine-tuning of LLMs to address privacy and resource constraints. GSQ-Tuning [1] enables fully integer-based on-device LLM fine-tuning using group-shared exponents, while QEFT [2] improves efficiency by updating only sensitive weight columns in mixed precision. These methods extend PEFT methods like LoRA [3] and QLoRA [4] for resource-constrained settings, and are applicable to cross-silo fine tuning. While our study focuses on cross-silo fine-tuning, we acknowledge that recent advances in on-device fine-tuning offer promising directions that could further enhance privacy and efficiency in federated settings, and we leave their exploration to future work.
> > > > >
> > > > > [1] GSQ-Tuning: Group-Shared Exponents Integer in Fully Quantized Training for LLMs On-Device Fine-tuning, ACL 2025.
> > > > > [2] QEFT: Quantization for Efficient Fine-Tuning of LLMs, ACL 2024.
> > > > > [3] LoRA: Low-rank adaptation of large language models, ICLR 2022.
> > > > > [4] QLoRA: Efficient finetuning of quantized llms, NeurIPS 2023.
> > > > > ```
> > > > >
> > > > > If the reviewer has any further questions or concerns, we would be more than happy to discuss them in detail.

---

> > > > > > ### Comment · Reviewer_ojVP · 2025-08-06
> > > > > >
> > > > > > Thanks for your detailed discussion. The rebuttal have solved my concerns.
> > > > > > Please add those discussion in related work and above rebuttal details in final revision. I recommand the positive for this paper and will raise my score to accept.

---

### Decision · Program_Chairs · 2025-09-18

**Decision:**

Accept (poster)

**Comment:**

This paper introduces FlowerTune, the first large-scale, cross-domain benchmark for federated fine-tuning of LLMs across four sensitive domains (general NLP, finance, medical, coding), with standardized datasets, evaluation metrics, and an open-source leaderboard. It offers a comprehensive empirical comparison of 26 models with multiple aggregation and PEFT strategies, yielding actionable insights into performance, resource constraints, and domain adaptation. While some limitations exist (e.g., uniform hyperparameters, limited FL scenarios), the authors have addressed concerns through rebuttal clarifications and added analyses. Overall, the benchmark is timely, impactful, reproducible, and community-driven, laying a strong foundation for privacy-preserving, domain-specialized LLMs. I recommend acceptance.

===== FINAL UPDATE FROM DB Track PCs ====

The final decision for this paper has been taken by the program chairs after consultation with the SACs. All Senior Area Chairs have ranked papers according to the feedback from the AC during the review process. We decided to leave the original meta-review to reflect the opinion of the AC in light of the initial discussions with reviewers and SAC.